# APPROXIMATED ANOMALOUS DIFFUSION: GAUSSIAN MIXTURE SCORE-BASED GENERATIVE MODELS

## ABSTRACT

Score-based generative models (SGMs) can generate high-quality samples via *Langevin dynamics* with a drift term and a diffusion term (Gaussian noise) iteratively calculated and added to a sample until convergence. In biological systems, it is observed that the neural population can conduct heavy-tailed *Lévy dynamics* for sampling-based probabilistic representation through neural fluctuations. Critically, unlike the existing sampling process of SGMs, heavy-tailed Lévy dynamics can produce both *large jumps* and *small roaming* to explore the sampling space, resulting in better sampling results than Langevin dynamics with a lacking of large jumps. Motivated by this contrast, we explore a new class of SGMs with the sampling based on the Lévy dynamics. However, exact numerical simulation of the Lévy dynamics is significantly more challenging and intractable. We hence propose an alternative solution by leveraging Gaussian mixture noises during training to mimic the desired large jumps and small roaming properties. Theoretically, GM-SGMs conduct a probabilistic graphical model used by empirical Bayes for sampling, expanding the maximum a posteriori (MAP) estimation applied by conventional SGMs. Expensive experiments on the challenging image generation tasks show that our GM-SGMs exhibit superior sampling quality over prior art SGMs across various sampling iterations.

## 1 INTRODUCTION

Score-based generative models (SGMs) (Song and Ermon, 2019; 2020; Song et al., 2021b;a; Dockhorn et al., 2022; Karras et al., 2022) have recently demonstrated tremendous performance in data synthesis, especially high-quality images, along with easier model optimization (Song and Ermon, 2019), richer generative diversity (Xiao et al., 2022), and solid theories (De Bortoli et al., 2021). During optimization, SGMs learn to fit a score function by predicting the Gaussian noises added to a sample drawn from a target dataset. To generate a sample in the target distribution, the SGMs conduct **Langevin dynamics** constructed from the score function. This process reverses a Brownian motion starting from the dataset distribution with *i.i.d.* Gaussian increments.

As a special case of Monte Carlo Markov Chain methods, the Langevin dynamics has been widely applied for constructing sampling-based algorithms (Rey-Bellet and Spiliopoulos, 2015). However, increasing evidence from experimental observation suggests that in biological systems, the neural population implements sampling-based probabilistic representations through a heavy-tailed Lévy dynamics (He, 2014; Donoghue et al., 2020; Townsend and Gong, 2018; Muller et al., 2018), which instead reverses an **anomalous diffusion process** with *heavy-tailed* increments (Fig. 2(Left)). The neural coding benefits from Lévy dynamics, since it can implement large jumps that facilitates the process to escape from local minimal and explore the sampling space more thoroughly (Ye and Zhu, 2018; Qi and Gong, 2022). A natural question arises: can we apply the Lévy dynamics instead of the Langevin dynamics for better sampling performance of SGMs?

Inspired by this insight, we explore a novel class of SGMs that reverse the anomalous diffusion for sampling. Nonetheless, exact numerical simulation of the Lévy dynamics (*i.e.*, reversing the anomalous diffusion) is drastically more challenging and intractable, especially for high-dimensional data such as images. To tackle this challenge, we consider *Brownian motion with Gaussian mixture* as an approximation of the anomalous diffusion. To tackle this challenge, we train the SGMs with *Gaussian mixture* noises to enable both large jumps and small roaming during the sampling phase,

reminiscent of the Lévy dynamics. Concretely, we construct a novel variant of SGMs, namely **Gaussian Mixture SGMs** (GM-SGMs), that learn to denoise Gaussian mixture noises; In doing so, our model is enabled to reverse the Gaussian mixture Brownian motion by switching between large jump and small roaming during sampling, resembling the merits of Lévy dynamics. Theoretically, our GM-SGMs conduct the probabilistic graphical model (PGM) of an empirical Bayes (EB) to reverse a Gaussian mixture Brownian motion; Instead, conventional SGMs perform a PGM of maximum a posteriori (MAP) estimation to reverse a Brownian motion. Empirically, extensive experiments on several challenging image generation tasks verify the ability of our GM-SGMs to automatically select large jump or small roaming during sampling and the promising ability of our GM-SGMs over state-of-the-art SGMs under different sampling budgets.

## 2 PRELIMINARY: SCORE-BASED GENERATIVE MODELS

Score matching was originally developed for non-normalized statistical learning (Hyvärinen and Dayan, 2005). By observing *i.i.d.* samples of an unknown (target) distribution $p^*$ in $d$ dimensions, score matching directly approximates the *score function* $\mathbf{s}(\mathbf{x}) := \nabla_{\mathbf{x}} \log p^*(\mathbf{x})$ via a model $\mathbf{s}_\theta$ parameterized by $\theta$, for $\mathbf{x} \in \mathbb{R}^d$. Score-based generative models (SGMs) aim to generate samples in the distribution $p^*$ via score matching through the following iterations

$$\mathbf{x}_T \sim \mathcal{N}(\mathbf{0}, I), \mathbf{x}_{t-1} = \mathbf{x}_t + \frac{\epsilon_t^2}{2}\mathbf{s}_\theta(\mathbf{x}_t, t) + \epsilon_t \mathbf{z}_t, \quad t = T, \dots, 1, \tag{1}$$

where $\epsilon_t$ is the step size and $\mathbf{z}_t \sim_{i.i.d.} \mathcal{N}(\mathbf{0}, I)$. This process transforms a Gaussian noise $\mathbf{x}_T$ towards a sample $\mathbf{x}_0$ obeying $p^*$. Eq.(1) can be considered as the reverse of a corrupting process where the noises are gradually added to a datum $\mathbf{x}_0$

$$\mathbf{x}_0 \sim p^*, \mathbf{x}_{t+1} = \mathbf{x}_t + \epsilon_t \mathbf{z}_t, \quad t = 0, \dots, T-1, \tag{2}$$

The first SGM, noise conditional score network (NCSN) (Song and Ermon, 2019), is trained by fitting the score function $\mathbf{s}(\mathbf{x})$ via minimizing the weighted *explicit score matching* (ESM) objective

$$L(\theta; \{\sigma_t\}_{t=1}^T) = \sum_{t=1}^T \lambda(\sigma_t) \mathbb{E}_{\mathbf{x} \sim p^*, \eta \sim \mathcal{N}(\mathbf{0}, \sigma_t^2 I)} \left[ \frac{1}{2} \|\mathbf{s}_\theta(\mathbf{x} + \eta, t) - \mathbf{s}(\mathbf{x} + \eta)\|^2 \right],$$

where $\sigma_t^2$ is the noise variance at the time step $t$, and $\lambda(\sigma_t)$ the weights for each time step $t$. Discarding the constant part independent from $\theta$, the ESM can be rewritten as a tractable *denoising score matching* (DSM) objective (Vincent, 2011)

$$L(\theta; \{\sigma_t\}_{t=1}^T) = \sum_{t=1}^T \lambda(\sigma_t) \mathbb{E}_{\mathbf{x} \sim p^*, \eta \sim \mathcal{N}(\mathbf{0}, \sigma_t^2 I)} \left[ \frac{1}{2} \|\mathbf{s}_\theta(\mathbf{x} + \eta, t) - \nabla_{\mathbf{x}+\eta} \log p_{\sigma_t}(\mathbf{x} + \eta \mid \mathbf{x})\|^2 \right]$$

$$= \sum_{t=1}^T \lambda(\sigma_t) \mathbb{E}_{\mathbf{x} \sim p^*, \eta \sim \mathcal{N}(\mathbf{0}, \sigma_t^2 I)} \left[ \frac{1}{2} \left\| \mathbf{s}_\theta(\mathbf{x} + \eta, t) + \frac{\eta}{\sigma_t^2} \right\|^2 \right], \tag{3}$$

where $p_{\sigma_t}(\cdot \mid \mathbf{x}) := \mathcal{N}(\cdot; \mathbf{x}, \sigma_t^2 I)$. Considering the sampling process as a stochastic differential equation (SDE), Song et al. (2021b) further proposed an improved version NCSN++ that utilizes an existing numerical solver of SDEs to enhance the sampling quality.

## 3 GAUSSIAN MIXTURE SGMS

### 3.1 ANOMALOUS DIFFUSION PERSPECTIVE

In this section, we introduce a wider class of corrupting processes as well as their reverses (*i.e.*, the corresponding sampling process), followed by analyzing their fundamental properties. We first write Eq.(1-2) in a continuous formula. Eq.(2) is the discretized version of a Brownian motion

$$\mathbf{x}_0 \sim p^*, d\mathbf{x} = d\mathbf{w}, \tag{4}$$

where the stochastic increment is Gaussian and satisfies

$$\Delta \mathbf{w}_t = \mathbf{w}_{t+\Delta t} - \mathbf{w}_t \sim \mathcal{N}(\mathbf{0}, \Delta t I), t, \Delta t > 0.$$

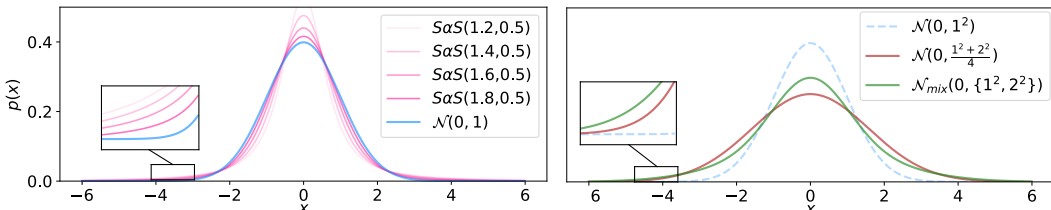

Figure 1: (**Left**) The pdf of one-dimensional Lévy stable distribution with different $\alpha$ and standard Gaussian distribution. The Lévy stable distribution is more heavy-tailed than the Gaussian distribution. (**Right**) Addition of two Gaussian distributions *vs.* their mixture. Best viewed in color.

The sampling process of SGMs (Eq.(1)) is the dicretized version of a Langevin dynamics

$$dx = \frac{1}{2}\epsilon^2(t) \bigtriangledown_x \log p^*(x)dt + \epsilon(t)d\bar{w}, \tag{5}$$

where $\epsilon(t)$ is a time-dependent positive scalar function, and $\bar{\cdot}$ represents reversing the time flow of the process. The Langevin dynamics (Eq.(5)) is the reverse process of the Brownian motion (Eq.(4)).

Further, Eq.(4) is an instance of a more general class of stochastic process called stable Lévy process,

$$x_0 \sim p^*, dx = dL^\alpha, \tag{6}$$

where $\alpha$ is the Lévy index, and the increment $\Delta L_t^\alpha$ obeys a Lévy stable distribution

$$\Delta L_t^\alpha = L_{t+\Delta t}^\alpha - L_t^\alpha \sim S\alpha S(x; \alpha, \Delta t^{\frac{1}{\alpha}}).$$

The Lévy stable distribution $S\alpha S$ has the probabilistic distribution function (pdf)

$$S\alpha S(x; \alpha, \gamma) = \frac{1}{\pi^d} \int_{\mathbb{R}_+^d} \exp\{-\frac{1}{2} \|\gamma y\|^\alpha\} \cos(x \cdot y)dy.$$

Similar as the Langevin dynamics (Eq.(5)), Eq.(6) has its reverse process called Lévy dynamics

$$dx = \frac{\epsilon(t)^2}{2p^*(x)}\mathcal{D}^{\alpha-2}\{p^*(x) \bigtriangledown_x \log p^*(x)\}dt + \epsilon(t)^{\frac{2}{\alpha}} d\bar{L}^\alpha, \tag{7}$$

where $\mathcal{D}^{\alpha-2}\{\cdot\}$ is the Riesz fractional derivative of order $\alpha$, and $dL^\alpha$ represents another stable Lévy process (Mandelbrot and Mandelbrot, 1982). Theoretically, we can sample from $p^*$ through the Lévy dynamics (Eq.(7)) that reverses a stable Lévy process (Eq.(6)).

When $\alpha = 2, \gamma = \frac{1}{2}$, $S\alpha S$ becomes the standard Gaussian distribution, and the stable Lévy process (Eq.(6)) becomes a Brownian motion. When $1 \leq \alpha < 2$, $S\alpha S$ is heavy-tailed, and the stable Lévy process becomes an anomalous diffusion (*a.k.a.*, super-diffusion or Lévy flight) (Klages et al., 2008; Metzler et al., 2014). To visually demonstrate the differences between Gaussian distribution and heavy-tailed Lévy stable distribution, we depict their probability distribution functions (pdf) in one-dimensional case. It is observed from Fig. 1 (left) that compared to Gaussian distribution, the Lévy stable distribution is not only more heavy-tailed but also more concentrated around zero center. In particular, the latter property prevents us from simply using a Gaussian distribution with high variance as an substitution of the Lévy stable distribution.

We further inspect the stochastic behavior differences of Brownian motion and anomalous diffusion resulted from their different distributions in the increment. As shown in Fig. 2, it is evident that the heavy-tailed property of Lévy stable distribution enables the sampling process (*i.e.*, the Lévy dynamics Eq.(7)) to better explore the full space more efficiently (Ye and Zhu, 2018). More specifically, the Lévy dynamics can produce large jumps for allowing the sample point to traverse though low-probability regions. As a result, it is less possible for the sample point to be trapped by local minimal. Also, it is easier for the sample point to switch between different modes, making it suitable to deal with multi-mode distribution situations (Qi and Gong, 2022). On the other hand, the higher concentration near zero eases the Lévy dynamics to conduct small roaming for exploring local regions. Consequently, the anomalous diffusion involves at least two modes: (1) *small roaming mode* for careful searching a local area, and (2) *large jump mode* for escaping from one area to a

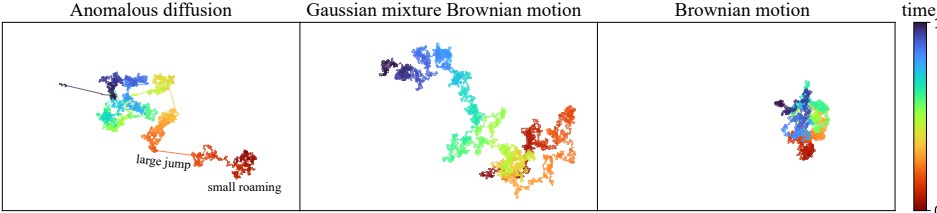

Figure 2: Visualization of (**Left**) anomalous diffusion $dL^\alpha$ ($\alpha = 1.8$), (**Middle**) Gaussian mixture Brownian motion $d\mathbf{w}^{1,\beta}$ ($\beta = 2$), and (**Right**) Brownian motion $d\mathbf{w}$. We fixed the random seed at 2023 for all the three simulations. Best viewed in color.

new area for global exploring. It has been widely observed in the neural population of biological systems, where the neural fluctuation is used for probabilistic coding (He, 2014; Donoghue et al., 2020; Townsend and Gong, 2018; Muller et al., 2018).

In contrast, the Gaussian distribution is relatively light-tailed, resulting in a lacking of large jumps in the Langevin dynamics (Eq.(5)). The sample point is thus easier to be trapped at local minimal. More importantly, for conventional SGMs, the light-tailed issue is not only caused by the stochastic increments $\mathbf{z}_t$, but also intensified by the model $\mathbf{s}_\theta$ itself, which plays the main role to guide the sample point to the target distribution $p^*$. To illustrate this, we revisit the score function in the conventional training objective (Eq.(3)), where the model $\mathbf{s}_\theta(\cdot, t)$ is trained to fit

$$\log p^*_{\sigma_t}(\mathbf{x} + \eta \mid \mathbf{x}) = -\frac{\eta}{\sigma_t^2}, \quad \mathbf{x} \sim p^*, \quad \forall \eta \in \mathbb{R}^d.$$

Ideally, the model $\mathbf{s}_\theta(\cdot, t)$ could learn the above map for **all** $\eta \in \mathbb{R}^d$, since the range of a Gaussian distribution is the whole $\mathbb{R}^d$. In practice, however, often $\mathbf{s}_\theta(\cdot, t)$ only learns this map well for those $\eta$ with relatively high probability being sampled from $\mathcal{N}(\mathbf{0}, \sigma_t^2 I)$, whilst lacking experiences for the low-probability others (*corresponding to the large jumps*). As a result, the conventional training only allows SGMs learn to reverse a Brownian motion, with less possible large jumps and inferior capability of predicting the corresponding samples.

This insightful analysis as above motivates us to design a new class of SGMs capable of conducting large jumps similar as the anomalous diffusion for sampling. As the reverse capability of a SGM is largely shaped by the distribution of the training noises $\eta$, the key is on reformulating this noise distribution for training.

### 3.2 GAUSSIAN MIXTURE

Although the anomalous diffusion comes with desired merits as discussed above, conducting a general Lévy dynamics (Eq.(7)) is computationally more challenging. *First*, the Riesz fractional derivative $\mathcal{D}\{\cdot\}$ is difficult to calculate (Çelik and Duman, 2012). *Second*, the Lévy dynamics involves $p^*$ which is intractable for high-dimensional cases. *Third*, sampling from a Lévy stable distribution (Mantegna, 1994) is much more expensive than sampling from a Gaussian distribution.

To bypass these obstacles, instead of struggling to approximate the Lévy dynamics, we resort to an alternative approach that mimics the functional properties (*i.e.*, ability to implement both large jumps and small roaming) of the anomalous diffusion. Formally, we exploit a Gaussian mixture sampling

$$\mathcal{N}_{\text{mix}}(0, \{I, \beta^2 I\}) := \frac{1}{2}\big(\mathcal{N}(0, I) + \mathcal{N}(0, \beta^2 I)\big),$$

where we mix $\mathcal{N}(0, I)$ with another Gaussian distribution with large variance ($\mathcal{N}(0, \beta^2 I), \beta > 1$). Note, our mixture operation ***sums two pdfs up to a single mixed pdf***, instead of taking the means of samples from two individual distributions respectively. We visualize their differences in Fig. 1 (Right): (1) The latter has much less concentration near zero. (2) Whilst the former (ours) largely keeps the original concentration degree, and is also more heavy-tailed than the latter.

Correspondingly, we define the *Gaussian mixture Brownian motion* as $d\mathbf{w}^{1,\beta}$, where the increment

$$\Delta\mathbf{w}_t^{1,\beta} = \mathbf{w}_{t+\Delta t}^{1,\beta} - \mathbf{w}_t^{1,\beta} \tag{8}$$

obeys the Gaussian mixture distribution $\mathcal{N}_{\text{mix}}(0, \{\Delta t I, \beta^2 \Delta t I\})$. To verify that our Gaussian mixture Brownian motion captures the behavior of the anomalous diffusion $dL^\alpha$, we conduct a two-dimensional simulation as shown in Fig. 2 (Middle). We can see that the large jumps property can be approximated well although not exactly for global space exploration, whilst preserving the small roaming trait in local regions.

## 3.3 TRAINING

As analyzed in Sec. 3.1, the distribution of $\eta$ in Eq.(3) determines what kinds of corrupting processes the SGM can reverse. To enable the model to reverse a Gaussian mixture Brownian motion, we propose **Gaussian mixture SGMs** (GM-SGM), which is trained to minimize

$$\sum_{t=1}^{T} \lambda(\sigma_t) \mathbb{E}_{\mathbf{x} \sim p^*(\mathbf{x}), \eta \sim \mathcal{N}_{\text{mix}}(\mathbf{0}, \{\sigma_t^2 I, \beta^2 \sigma_t^2 I\})} \left[ \frac{1}{2} \left\| \mathbf{s}_\theta(\mathbf{x} + \eta, t) + \frac{\eta}{\sigma_t^2} \right\|^2 \right], \qquad (9)$$

where $\beta \geq 1$ is the only hyper-parameter. Compared to the conventional training objective (Eq.(3)), the only difference is that the noise $\eta$ now is sampled from the Gaussian mixture $\mathcal{N}(\mathbf{0}, \{\sigma_t^2 I, \beta^2 \sigma_t^2 I\})$ instead of $\mathcal{N}(\mathbf{0}, \sigma_t^2 I)$ at each time step $t$. Minimizing our objective Eq.(9) is equivalent to alternatively minimize the original objective Eq.(3) and the following objective

$$\sum_{t=1}^{T} \lambda(\sigma_t) \mathbb{E}_{\mathbf{x} \sim p^*(\mathbf{x}), \eta \sim \mathcal{N}(\mathbf{0}, \sigma_t^2 I)} \left[ \frac{1}{2} \left\| \mathbf{s}_\theta(\mathbf{x} + \beta\eta, t) + \frac{\beta\eta}{\sigma_t^2} \right\|^2 \right]. \qquad (10)$$

This provides new opportunities, where the scaled noises $\beta\eta$ added to the sample $\mathbf{x}$ have much larger deviation than expected, forcing the model conduct a *large jump* in order to recover $\mathbf{x} \sim p^*$.

As there is no indication with the input about if the noise $\eta$ is scaled by $\beta$ or not, the model must *learn to decide on its own* when to make the large jump, *i.e.*, learning to switch between large jump mode and small roaming mode properly. After training, at each time step $t$, our GM-SGM will automatically select one of possible denoising prior: the noise to be denoised at this step obeys $\mathcal{N}(\mathbf{0}, \sigma_t^2 I)$ or $\mathcal{N}(\mathbf{0}, \sigma_t^2 \beta^2 I)$. This enables GM-SGM to vary the step size adaptively to some degree with more robustness against outlier cases (*e.g.*, $\eta$ satisfying $\mathcal{N}(\eta; \mathbf{0}, \sigma_t^2 I) \ll 1$).

To reduce the stochasticity of training, we alternatively minimize two objectives Eq.(3) and Eq.(10), instead of uniformly sampling them. The training procedure is summarized in Alg. 1.

---

**Algorithm 1** GM-SGM training

**Input:** The model $\mathbf{s}_\theta$, the overall training iterations $N$, the parameter $\beta$.
**for** $n = 1$ **to** $N$ **do**
    Draw $\mathbf{x} \sim p^*(\mathbf{x})$, $t \sim \text{Uniform}[1, 2, \ldots, T]$, $\eta \sim \mathcal{N}(\mathbf{0}, I)$
    **if** $n \mod 2 = 0$ **then**
        $\tilde{\beta} = 1$
    **else**
        $\tilde{\beta} = \beta$
    **end if**
    Backpropagate on $\lambda(\sigma_t) \left\| \mathbf{s}_\theta(\mathbf{x} + \tilde{\beta}\eta, t) + \frac{\tilde{\beta}\eta}{\sigma_t^2} \right\|^2$
**end for**
**Output:** $\mathbf{s}_\theta$

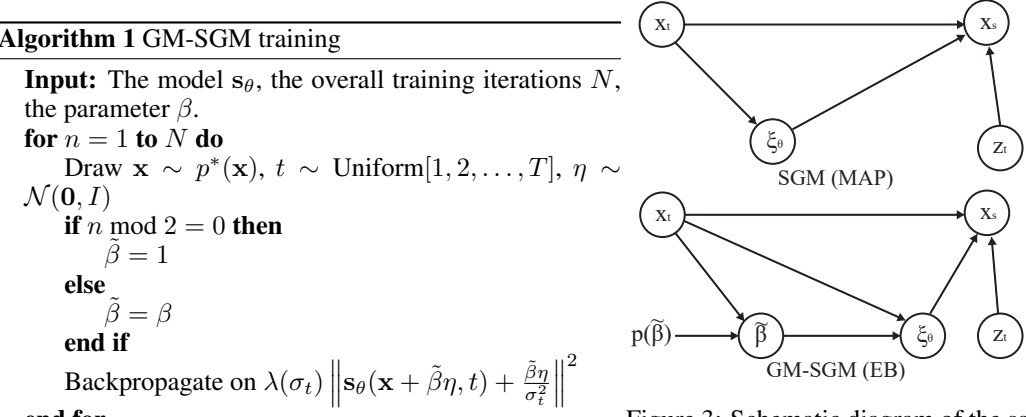

Figure 3: Schematic diagram of the sampling inference: SGMs *vs.* GM-SGMs.

---

## 3.4 SAMPLING

During inference, we sample from a trained GM-SGM using exactly the same procedure as conventional SGMs (Eq.(1)). Thus, our GM-SGMs do not introduce any extra cost during inference. This is some inconsistent with our training noise increments $\mathbf{z}_t \sim \mathcal{N}_{\text{mix}}(\mathbf{0}, \{I, \beta^2 I\})$. We provide the theoretical insights and analysis as follows.

## 4 THEORETICAL ANALYSIS

### 4.1 UNDERSTANDING GM-SGMS VIA PROBABILISTIC INFERENCE

GM-SGMs can be extended to general cases, where we train the model via the Gaussian mixture

$$\mathcal{N}_{\mathrm{mix}}(\mathbf{x}; \mathbf{0}, p(\tilde{\beta})) := \int \mathcal{N}(\mathbf{x}; \mathbf{0}, \tilde{\beta}^2 I) p(\tilde{\beta}) d\tilde{\beta},$$

where $p(\tilde{\beta})$ is a *possibly continuous pdf* of the scalar $\tilde{\beta}$. In Sec. 3.2, the distribution of $\tilde{\beta}$ is *discrete*: $p(\tilde{\beta} = 1) = p(\tilde{\beta} = \beta) = 0.5$, corresponding to the small roaming and large jump modes respectively. In this general case, the corresponding objective is

$$\sum_{t=1}^{T} \lambda(\sigma_t) \mathbb{E}_{\tilde{\beta} \sim p(\tilde{\beta})} \left[ \mathbb{E}_{\mathbf{x} \sim p^*(\mathbf{x}), \eta \sim \mathcal{N}(\mathbf{0}, \sigma_t^2 I)} \left[ \frac{1}{2} \left\| \mathbf{s}_\theta(\mathbf{x} + \tilde{\beta}\eta, t) + \frac{\tilde{\beta}\eta}{\sigma_t^2} \right\|^2 \Big| \tilde{\beta} \right] \right]. \tag{11}$$

To understand why the sampling process of GM-SGM can converge to the target distribution, we analyze what the GM-SGM is really doing from the perspective of probabilistic inference.

Let us first revisit the conventional SGMs. For convenience, we denote $-\sigma_t \mathbf{s}_\theta(\mathbf{x} + \sigma_t \xi, t)$ as $\xi_\theta(\mathbf{x}, t)$, an equivalent form of the SGM. Then $\xi_\theta$ is trained to infer the standardized noise $\xi$ through the probabilistic graphical model (PGM)

$$p(\xi \mid \mathbf{x}_t, t) \propto p(\mathbf{x}_t \mid \xi, t) p(\xi),$$

given $\mathbf{x}_t = \mathbf{x}_0 + \sigma_t \xi$ and $t$, where $\mathbf{x}_0 \sim p^*, \xi \sim \mathcal{N}(\mathbf{0}, I)$, and $p(\xi) = \mathcal{N}(\mathbf{0}, I)$. This PGM is the same as that in a *maximum a posteriori estimation* (MAP) formula. After estimating $\xi$ by $\xi_\theta$, at step $t$, the model can push the sample point $\mathbf{x}_t$ along the direction of $-\xi$ to approximate the target distribution. This approximation is reasonable due to following relationship.

**Theorem 1.** *Given the corrupting process* ($\sigma_t > \sigma_s, \forall t > s$)

$$\mathbf{x}_0 \sim p^*, \quad \mathbf{x}_t = \mathbf{x}_0 + \sigma_t \xi_t, \quad \xi_t \sim_{i.i.d.} \mathcal{N}(\mathbf{0}, I), \tag{12}$$

*for any time points* $0 \le u < s < t$*, we have*

$$p(\mathbf{x}_s \mid \mathbf{x}_t, \mathbf{x}_u) = \mathcal{N}\big(\mathbf{x}_t - \frac{\sigma_t^2 - \sigma_s^2}{\sigma_t^2 - \sigma_u^2}(\mathbf{x}_t - \mathbf{x}_u), \frac{(\sigma_t^2 - \sigma_s^2)(\sigma_s^2 - \sigma_u^2)}{\sigma_t^2 - \sigma_u^2} I\big).$$

See the proof in A.1 of `Appendix`. As a result, if we set $u = 0$, it is clear that if $\mathbf{x}_t$ moves along the direction $-\xi = -(\mathbf{x}_t - \mathbf{x}_0)/\sigma_t$ with small enough step, it will be closer to $\mathbf{x}_s$ for $s < t$. During sampling, the SGM takes the step along this direction with a scalar $\frac{\epsilon_t^2}{2}$ controlling the step size at every time step $t$ and a random increment $\mathbf{z}_t$ that represents the uncertainty in $p(\mathbf{x}_s \mid \mathbf{x}_t, \mathbf{x}_0)$.

When applying the above analysis to GM-SGM, we find that this time $\xi_\theta$ is trained to infer $\xi$ through

$$p(\xi \mid \mathbf{x}_t, t) \propto \int p(\mathbf{x}_t \mid \xi, t) p(\xi \mid \tilde{\beta}) p(\tilde{\beta} \mid \mathbf{x}_t, t) d\tilde{\beta}, \quad p(\tilde{\beta} \mid \mathbf{x}_t, t) \propto p(\mathbf{x}_t \mid \tilde{\beta}, t) p(\tilde{\beta}) \tag{13}$$

where $p(\tilde{\beta})$ is the distribution of $\tilde{\beta}$ in Eq.(11), and $p(\xi \mid \tilde{\beta}) = \mathcal{N}(\xi; \mathbf{0}, \tilde{\beta}^2 I)$. Therefore, GM-SGMs also infer a direction $-\xi$ to push the sample point $\mathbf{x}_t$ closer to $\mathbf{x}_s, s < t$. The key difference is that infers the direction $-\xi$ is inferred through the PGM of *maximum a posteriori estimation* (MAP) formula with the SGMs, whilst through the PGM of *empirical Bayes* (EB) formula (BP and TA, 2001) by our GM-SGM. As illustrated in Fig. 3, GM-SGM takes into consideration of the probability that $\xi$ is associated with different covariance matrices $\tilde{\beta}^2 I$, corresponding to large jumps and small roaming.

With the above analysis, we can more clearly explain why we apply the Langevin dynamics as the sampling process of GM-SGM, with the increments $\mathbf{z}_t$ in Gaussian distribution instead of Gaussian mixture. From the perspective of probabilistic inference, $\mathbf{z}_t$ represents ***the level of uncertainty of estimating*** $\xi_\theta$ ***at time step*** $t$. Indeed, during training, we do not specify different levels of uncertainty of $\xi$ for different $\tilde{\beta}$ given the time $t$. This means the model infers $\xi$ at the same level of certainty at each time step $t$, independent of $\tilde{\beta}$. Therefore, for GM-SGM we can adopt the sampling process as the SGMs. Otherwise, when Gaussian mixture increments are used for sampling, the model performance would degrade due to inconsistent uncertainty introduced (verified in Sec. 5).

## 4.2 RELATIONSHIP TO THE CONVENTIONAL SGMs

Following the relationships between our GM-SGMs and conventional SGMs as analyzed in Sec. 4.1, we further make additional interesting connections as follows.

First, GM-SGMs can be considered as a more robust variant of SGMs. Extra robustness comes from that the model is optimized to deal with large jumps $\tilde{\beta}\eta, \eta \sim \mathcal{N}(\mathbf{0}, \tilde{\beta}^2\sigma_t^2 I), \tilde{\beta} > 1$ (the outliers of $\mathcal{N}(\mathbf{0}, \sigma_t^2 I)$) at each step $t$ (Eq.(10)).

Second, training GM-SGMs can be regarded as an ensemble learning procedure of training SGMs. Specifically, for every $\tilde{\beta}$, we train a SGM (denoted as $\mathbf{s}_{\theta,\tilde{\beta}}$) by minimizing corresponding objective constructed by $\tilde{\beta}$, followed by mixing these SGMs based on $p(\tilde{\beta})$.

Third, the sampling process of GM-SGMs can be seen as reversing a mixture of multiple Brownian motions, where the noise term $\mathbf{z}_t$ is amplified by different $\tilde{\beta}$. These corrupting processes expand the hypothesis space of original SGMs which only contains a single Brownian motion. The model $\mathbf{s}_\theta$ adaptively selects one of them for denoising, *i.e.*, switching from different modes ($\tilde{\beta} > 1$ for large jumps and $\tilde{\beta} \leq 1$ for small roaming). This will be verified in Fig. 4. As a result, going beyond conventional SGMs with a need for designing a step schedule for sampling, our GM-SGMs can vary the step sizes of the score function term *automatically*.

## 5 EXPERIMENTS

We evaluate image generation tasks on CIFAR-10 (Krizhevsky et al., 2009), CelebA (Liu et al., 2015) and LSUN (church and bedroom) (Yu et al., 2015). We focus on comparing GM-SGMs with the prior art SGMs, namely NCSN++ (Song et al., 2021b). For fair comparison, we construct our proposed GM-SGM using the same U-Net architecture as NCSN++. We use the released checkpoints of NCSN++ for all the datasets except CelebA for which no checkpoints released and we use the released codes to train by ourselves. We first set $\beta = 2$ in Alg. 1 and investigate the effect of different $\beta$ later. We provide more detailed settings of implementation in A.2 of `Appendix`.

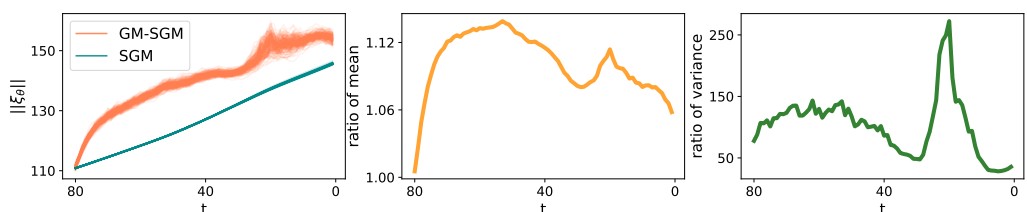

Figure 4: The norm $\|\xi_\theta\|$ of GM-SGMs and SGMs. (***Left***) The norms across several sampling processes. The ratio of (***Middle***) mean $r_{\text{mean}}$ and (***Right***) variance $r_{\text{var}}$ at different time steps.

### 5.1 SAMPLING STATISTICS

To verify that our GM-SGM indeed makes large jumps following the PGM of empirical Bayes inference, we track the values of the score function $\mathbf{s}_\theta(\mathbf{x}_t, t)$ evaluated during the sampling procedure and calculate the norm of the estimated noise vector $\xi_\theta(\mathbf{x}_t, t)$ for every time step $t$ (see Sec. 4.1). We denote $\xi_\theta^{\text{GM-SGM}}(\mathbf{x}_t, t)$ and $\xi_\theta^{\text{SGM}}(\mathbf{x}_t, t)$ as the noise vector produced by our GM-SGM and SGMs respectively. For a relative comparison, we calculate the ratio of mean and the ratio of variance

$$r_{\text{mean}}(t) = \frac{\mathbb{E}\left[\left\|\xi_\theta^{\text{GM-SGM}}(\mathbf{x}_t, t)\right\|\right]}{\mathbb{E}\left[\left\|\xi_\theta^{\text{SGM}}(\mathbf{x}_t, t)\right\|\right]}, \quad r_{\text{var}}(t) = \frac{\text{var}\left[\left\|\xi_\theta^{\text{GM-SGM}}(\mathbf{x}_t, t)\right\|\right]}{\text{var}\left[\left\|\xi_\theta^{\text{SGM}}(\mathbf{x}_t, t)\right\|\right]},$$

where $r_{\text{mean}}/r_{\text{var}}$ represent the relative denoising step-size/variation taken by our GM-SGM over convnetional SGMs on average. We sample 512 images using GM-SGM and SGM trained on CelebA ($64 \times 64$) (Liu et al., 2015) respectively and plot the results of sampling statistics in Fig. 4. We draw several key observations: (**1**) With the $r_{\text{mean}} > 1$ across the whole sampling process, we validate that our GM-SGM does take more large jumps. (**2**) $r_{\text{var}}$ starts at the range of $[50, 150]$ and surges

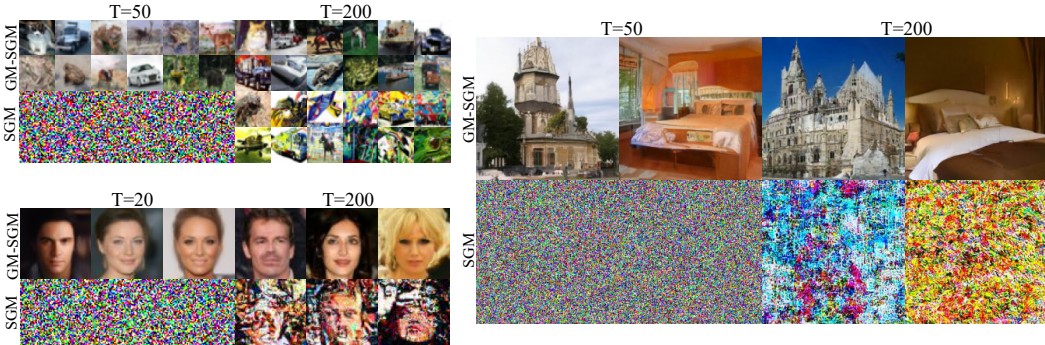

Figure 5: Image samples generated by SGMs and GM-SGMs on CIFAR-10, CelebA with $64 \times 64$ resolution and LSUN (church and bedroom) with $256 \times 256$ resolutions.

rapidly afterwards followed by a quick dip. This implies that when the sample point is sufficiently informative (*e.g.*, close to the target distribution), our GM-SGM would activately infer the noises $\xi$ based on different hypothesis of $\tilde{\beta}$. In contrast, the norm variation by SGM is always much smaller. The underlying reason is that, GM-SGM can take much more diverse sampling paths, as expressed by Eq.(13). More specifically, at every step $t$, GM-SGM estimate $\xi$ with two possible selections in the prior distribution of $\xi$: $\mathcal{N}(0, I)$ or $\mathcal{N}(0, \beta^2 I)$ at each $t$. Therefore, there are a total of $2^T$ denoising paths GM-SGM can choose from during sampling.

## 5.2 Evaluation on image synthesis quality

Next, we evaluate the sampling quality. As shown in Fig. 5, our GM-SGMs can generate much better quality images than SGMs under the same iterations $T$. See A.7 of `Appendix` for more visual comparison. For quantitative measurement, we adopt the popular Fréchet inception distance (FID) (Heusel et al., 2017). Lower FID is better. As shown in Table 1-2, GM-SGM is consistently superior over SGM across different sampling budgets and generation tasks.

Table 1: FID score of generating object images on CIFAR-10. Lower FID is better. SGM: NCSN++ Song et al. (2021b)).

| Iterations $T$ | SGM | GM-SGM |
|---|---|---|
| 1000 | 2.50 | **2.25** |
| 500 | 3.54 | **3.53** |
| 400 | 4.35 | **4.00** |
| 333 | 6.24 | **4.55** |
| 250 | 12.70 | **6.18** |
| 200 | 29.39 | **8.76** |
| 100 | 306.91 | **37.63** |
| 50 | 456.49 | **29.41** |

Table 2: FID score of generating human facial images on CelebA ($64 \times 64$). Lower FID is better. SGM: NCSN++ Song et al. (2021b)).

| Iterations $T$ | SGM | GM-SGM |
|---|---|---|
| 500 | 4.01 | **3.75** |
| 400 | 6.56 | **3.98** |
| 333 | 14.53 | **4.10** |
| 200 | 257.64 | **4.51** |
| 100 | 435.59 | **5.99** |
| 80 | 437.98 | **6.20** |
| 66 | 436.91 | **6.26** |
| 20 | 439.32 | **27.81** |

## 5.3 GM-SGM parameter analysis

We investigate the effect of $\beta$ on sampling quality. The default setting is $\beta = 2$. In general, if $\beta$ is over small, our model advantage will be reduced due to reduced heavy-tail property. If $\beta$ is over large, the denoising task would become too challenge to solve, hence harming the performance. To empirically validate this, we test GM-SGMs by varying $\beta$ in $[1.5, 3]$ on CIFAR-10 with 50 iterations. When $\beta = 1.5$, the FID increases from 29.4 to 37.9. When $\beta = 3$, the FID goes to 105.4. However, both are still better than 456.5 by SGM. See Tab. 3 in A.3 of `Appendix` for more results. Also, we test other Gaussian mixture designs and find them less effective (see A.6 of `Appendix`).

### 5.4 FURTHER STUDY

We investigate more designs of GM-SGMs. *First*, we only use the objective Eq.(10) for training. As shown in Fig. 7 of `Appendix`, the model fails to generate natural images even under many iterations. This is because this model only learns to implement large jumps, while ignoring small roaming which is also necessary for space exploring, as discussed in Sec. 3.1. *Second*, instead of Gaussian increments as SGMs, we turn to use Gaussian mixture $\mathcal{N}(0, \{I, \beta^2 I\})$. We test this on CelebA ($64 \times 64$) under 333 iterations. We find the FID increases from $4.1$ to $5.2$ (still much better than $14.5$ by SGMs). This verifies empirically our analysis in Sec. 4.1 that the probabilistic inference of our GM-SGM shares the same level of uncertainty for all $\beta$ at each step $t$. We also investigate GM-SGMs using Gaussian mixture with general covariance matrices in A.5 of `Appendix`.

## 6 RELATED WORK

The first SGMs, the noise conditional score network (NCSN), is introduced (Song and Ermon, 2019). Later on, Song and Ermon (2020) further improved the NCSN by scaling the noises and improving the stability with the moving average. By abstracting the previous SGMs into a unified framework based on the stochastic differential equation (SDE), Song et al. (2021b) proposed the NCSN++ for high-resolution image generation using numerical SDE solvers and several architectural enhancements. Relying on Hamiltonian Monte Carlo methods (Neal et al., 2011), critically-damped Langevin diffusion (CLD) based SGMs is introduced (Dockhorn et al., 2022). Alternatively, Jing et al. (2022) implemented the sampling process on a series of selected subspaces. Besides, Vahdat et al. (2021) trained the SGMs in a latent space of variational autoencoder (Kingma and Welling, 2019). While Karras et al. (2022) focused on refining the training hyper-parameters, sampling schedule and high-order numerical methods for better performance.

DDPMs (Ho et al., 2020) are another class of iterative denoising based generation models. Their origin is (Sohl-Dickstein et al., 2015) that proposed to destroy the data through a diffusion process, and learn to reverse this process via maximizing the variational bound. Although theoretically equivalent to SGMs (Vincent, 2011), DDPMs adopt a variance preserving (VP) stochastic process for sampling. In contrast, SGMs consider a variance exploding (VE) stochastic process (*i.e.*, the Langevin dynamics). There have been further improvements for DDPMs (Yang et al., 2022). For instance, DDIMs (Song et al., 2020) generate samples via a class of non-Markovian diffusion processes. Nichol and Dhariwal (2021) proposed to learn the noise schedules for better sampling quality. Liu et al. (2021) further improved DDIMs by implementing the sampling on a manifold for better denoising. Bao et al. (2022b;a) attempted to optimize the discrete-time schedules for sampling speed-up.

Application of SGMs and DDPMs has been rapidly evolved, *e.g.*, text-to-image generation (Nichol et al., 2021; Saharia et al., 2022; Rombach et al., 2022). Recently, SGMs were applied in creating, editing, and recovering photo-realistic images (Meng et al., 2021; Saharia et al., 2021a;b; Kawar et al., 2022), or high fidelity audio streams (Kong et al., 2021; Chen et al., 2021).

Complementary to all the prior efforts, in this work we investigate the fundamental limitation of lacking large jumps in sampling with conventional SGMs and construct a more robust variant characterized by superior exploration ability, along with solid theoretical justification.

## 7 CONCLUSION

Inspired by the heavy-tailed Lévy dynamics (the reverse of the anomalous diffusion) implemented by neural coding in biological systems, we propose a novel class of generative models, namely *Gaussian mixture score-based generative models* (GM-SGMs). In contrast to conventional SGMs based on Langevin dynamics, our models are featured with a relatively heavy-tailed sampling property and capability of more thoroughly exploring the sampling space. Specifically, GM-SGMs are trained by denoising Gaussian mixture noises with built-in automatic switches between large jumps and small roaming in the spirit of anomalous diffusion. Theoretically, GM-SGMs implement a probabilistic graphical model (PGM) of an empirical Bayesian for generation, expanding the PGM of maximum a posteriori estimation used by the SGMs; This results in a sampling process with automatic selection of the step schedule and more robustness. Empirically, we demonstrate that our GM-SGMs significantly outperform the conventional SGMs on a diversity of challenging image generation tasks.

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

# A APPENDIX

## A.1 PROOF OF THM. 1

*Proof.* Using Bayes' theorem, we have

$$p(\mathbf{x}_s \mid \mathbf{x}_t, \mathbf{x}_u) \propto p(\mathbf{x}_t \mid \mathbf{x}_s, \mathbf{x}_u) p(\mathbf{x}_s \mid \mathbf{x}_u).$$

According to Eq.(12), we have

$$p(\mathbf{x}_t \mid \mathbf{x}_s, \mathbf{x}_u) = p(\mathbf{x}_t \mid \mathbf{x}_s) = \frac{1}{\sqrt{2\pi(\sigma_t^2 - \sigma_s^2)}^d} \exp\{-\frac{1}{2}\frac{1}{\sigma_t^2 - \sigma_s^2}\|\mathbf{x}_t - \mathbf{x}_s\|^2\}$$

and

$$p(\mathbf{x}_s \mid \mathbf{x}_u) = \frac{1}{\sqrt{2\pi(\sigma_s^2 - \sigma_u^2)}^d} \exp\{-\frac{1}{2}\frac{1}{\sigma_s^2 - \sigma_u^2}\|\mathbf{x}_s - \mathbf{x}_u\|^2\}$$

After taking logarithm at the both sides of the equation and ignoring a constant term, we have

$$\log p(\mathbf{x}_s \mid \mathbf{x}_t, \mathbf{x}_u) = -\frac{1}{2}\frac{1}{\sigma_t^2 - \sigma_s^2}\|\mathbf{x}_t - \mathbf{x}_s\|^2 - \frac{1}{2}\frac{1}{\sigma_t^2 - \sigma_s^2}\|\mathbf{x}_t - \mathbf{x}_s\|^2$$

$$= -\frac{1}{2}\frac{\sigma_t^2 - \sigma_u^2}{(\sigma_t^2 - \sigma_s^2)(\sigma_s^2 - \sigma_u^2)}\left\|\mathbf{x}_s - \frac{(\sigma_t^2 - \sigma_s^2)(\sigma_s^2 - \sigma_u^2)}{\sigma_t^2 - \sigma_u^2}(\frac{\mathbf{x}_t}{\sigma_t^2 - \sigma_s^2} + \frac{\mathbf{x}_u}{\sigma_s^2 - \sigma_u^2})\right\|^2$$

$$= -\frac{1}{2}\frac{\sigma_t^2 - \sigma_u^2}{(\sigma_t^2 - \sigma_s^2)(\sigma_s^2 - \sigma_u^2)}\left\|\mathbf{x}_s - \left(\mathbf{x}_t - \frac{(\sigma_t^2 - \sigma_s^2)(\sigma_s^2 - \sigma_u^2)}{\sigma_t^2 - \sigma_u^2}\frac{1}{\sigma_s^2 - \sigma_u^2}(\mathbf{x}_t - \mathbf{x}_u)\right)\right\|^2$$

$$= -\frac{1}{2}\frac{\sigma_t^2 - \sigma_u^2}{(\sigma_t^2 - \sigma_s^2)(\sigma_s^2 - \sigma_u^2)}\left\|\mathbf{x}_s - \left(\mathbf{x}_t - \frac{\sigma_t^2 - \sigma_s^2}{\sigma_t^2 - \sigma_u^2}(\mathbf{x}_t - \mathbf{x}_u)\right)\right\|^2,$$

which implies that $p(\mathbf{x}_s \mid \mathbf{x}_t, \mathbf{x}_u)$ also obeys a Gaussian distribution, and its mean and covariance matrix can be obtained

$$\mathbb{E}_{\mathbf{x}_s \sim p(\mathbf{x}_s \mid \mathbf{x}_t, \mathbf{x}_u)}[\mathbf{x}_s] = \mathbf{x}_t - \frac{\sigma_t^2 - \sigma_s^2}{\sigma_t^2 - \sigma_u^2}(\mathbf{x}_t - \mathbf{x}_u),$$

$$\mathrm{cov}_{\mathbf{x}_s \sim p(\mathbf{x}_s \mid \mathbf{x}_t, \mathbf{x}_u)}[\mathbf{x}_s] = \frac{(\sigma_t^2 - \sigma_s^2)(\sigma_s^2 - \sigma_u^2)}{\sigma_t^2 - \sigma_u^2}I$$

$\square$

Hence, the estimated direction $-\xi_\theta(\mathbf{x}_t, t) = -(\mathbf{x}_t - \mathbf{x}_0)/\sigma_t$ guides the sample point $\mathbf{x}_t$ closer to $\mathbf{x}_s$ for some $s < t$ whose distribution is closer to $p^*$.

## A.2 DETAILED SETTINGS

We train the GM-SGM on CIFAR-10 for 550k iterations at the batch size of 64 on single NVIDIA RTX 3090 GPU to compare the public checkpoint of the SGM (NCSN++) trained on CIFAR-10. We train both GM-SGM and SGMs on CelebA ($64 \times 64$) for 230k iterations at the batch size of 64 on two NVIDIA RTX 3090 GPUs. We train the GM-SGM on LSUN (church) and LSUN (bedroom) at resolution of $256 \times 256$ for 250k iterations at the batch size of 64 and compare the public checkpoint of the SGM (NCSN++) trained on LSUN (church) on 4 NVIDIA RTX A6000 GPUs. We apply the same hyper-parameters of optimizer and learning schedule as that in Song et al. (2021b). For sampling, we use the same generation process of continuous variance exploding (VE)-NCSN++ Song et al. (2021b) with both predictor and corrector for both SGMs and GM-SGM. For sampling, all the experiments apply the original linear schedule used in Song et al. (2021b) with both predictor and corrector (PC).

## A.3 EXTRA RESULTS ON PARAMETER ANALYSIS

We report more results on how the parameter $\beta$ influences the performance of GM-SGM in Tab. 3. In particular, when $\beta = 1$, we get the original SGM.

Table 3: The comparison of the performance of GM-SGMs under different $\beta$ in Alg. 1. Metrics: FID score. Dataset: CIFAR-10.

| Iterations T | 50 | 200 | 250 | 500 |
|---|---|---|---|---|
| $\beta = 1$ (SGM) | 456.49 | 29.39 | 12.70 | 3.54 |
| $\beta = 1.5$ | 37.91 | 9.06 | 7.07 | **3.33** |
| $\beta = 2$ | **29.41** | **8.76** | **6.18** | 3.53 |
| $\beta = 3$ | 105.40 | 11.24 | 8.08 | 3.64 |

## A.4 Comparison to other methods

In this section, we compare our GM-SGMs with existing SGMs and DDPMs. First, we compare GM-SGMs with SGMs equipped with adaptive SDE solver (ASDE) (Jolicoeur-Martineau et al., 2021). As shown in Table 4, our GM-SGMs outperform the competitors significantly. This suggests that our method has superior acceleration benefit. Also note our GM-SGMs do not alter the sampling process, without adding extra per-step complexity during inference. In contrast, alternative acceleration methods usually increase per-step cost.

Table 4: FID of SGMs (Song et al., 2021b), GM-SGMs, and SGMs equipped with adaptive SDE solver (Jolicoeur-Martineau et al., 2021). Dataset: CIFAR-10.

| | adaptive SDE solver (VP) | adaptive SDE solver (VE) | SGM | **GM-SGM (ours)** |
|---|---|---|---|---|
| Iterations T | 49 | 50 | 50 | 50 |
| FID | 82.42 | 307.32 | 456.49 | **29.41** |

Next, we compare GM-SGMs with DDPMs using non-Gaussian noises for training or sampling, including DDGM (Nachmani et al., 2021) where Gaussian noises are replaced by noises obeying Gamma distribution. As shown in Table 5, our GM-SGMs are superior over DDPMs, DDPM-based DDGM, and comparable to DDIM and DDIM-based DDGM, despite that their base model DDPM and DDIM already have better low-iteration performance than our base model SGMs.

Table 5: FID of SGMs (Song et al., 2021b), GM-SGMs, DDPM (Ho et al., 2020), DDIM (Song et al., 2020), and DDGM (based on DDPM and DDIM) (Nachmani et al., 2021). Dataset: CelebA $(64 \times 64)$.

| Iterations T | DDPM | DDIM | DDGM (DDPM-based) | DDGM (DDIM-based) | SGM | **GM-SGM (ours)** |
|---|---|---|---|---|---|---|
| 20 | 183.83 | 13.73 | 28.24 | 6.83 | 439.32 | 27.81 |
| 100 | 45.2 | 6.53 | 14.22 | 3.17 | 435.59 | 5.99 |

## A.5 More general cases

In Sec. 4.1, we consider the cases of using Gaussian mixture $\mathcal{N}_{\text{mix}}(\mathbf{x}; \mathbf{0}, p(\tilde{\beta}))$ for training. In fact, this can be generalized to multi-dimensional Gaussian mixture cases. Denoting a distribution of *invertible matrix* $\tilde{B}$ as $p(\tilde{B})$, we can consider a more general training objective

$$\sum_{t=1}^{T} \lambda(\sigma_t) \mathbb{E}_{\tilde{B} \sim p(\tilde{B})} \left[ \mathbb{E}_{\mathbf{x} \sim p^*, \eta \sim \mathcal{N}(\mathbf{0}, \sigma_t^2 I)} \left[ \frac{1}{2} \left\| \mathbf{s}_\theta(\mathbf{x} + \tilde{B}\eta, t) + \frac{\tilde{B}\eta}{\sigma_t^2} \right\|^2 \bigg| \tilde{B} \right] \right].$$

It is not obvious which kinds of $p(\tilde{B})$ can train a better GM-SGM. Here, we try a $p(\tilde{B})$ motivated by the observation that for image generation, SGMs tend to generate images with too much high-frequency noise when the iterations number is low (see examples in Sec. 3.4). To suppress the generation for high-frequency components, we design a $p(\tilde{B})$ matrix as follows

$$p(\tilde{B} = I) = 0.5, P(\tilde{B} = 2M) = 0.5,$$

where $M$ is a filter operated on the height and width coordinates to shrink the top-$5\%$ high-frequency part of an image by $0.5$. We obtain the following results: For CelebA ($64 \times 64$), the FID of GM-SGM decreases to 3.77 at 400 iterations (better than the result of GM-SGM in the main paper); On CIFAR-10, the FID of GM-SGM decreases to 2.24 at 1000 iterations, better than SGMs (2.51). This implies that the optimal settings of Gaussian mixture can be more complicated, which calls for a further study in the future.

### A.6   FAILURE EXAMPLES

In Fig. 6 and Fig. 7, we show the failure examples under improper parameter settings as mentioned in Sec. 5.

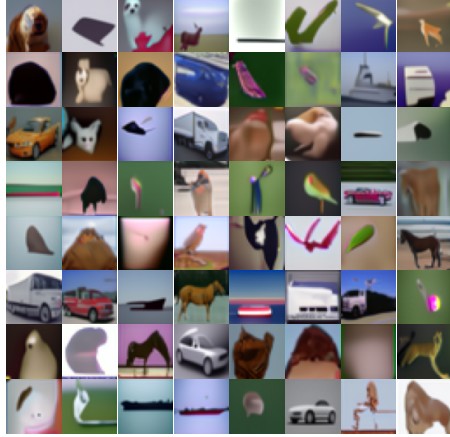

Figure 6: Image samples generated by a GM-SGM trained on CIFAR-10, where we apply the Gaussian mixture $\mathcal{N}(0, \{0.5^2 I, 2^2 I\})$ instead of Gaussian mixture $\mathcal{N}(0, \{I, 2^2 I\})$ for training. Iterations: 2000.

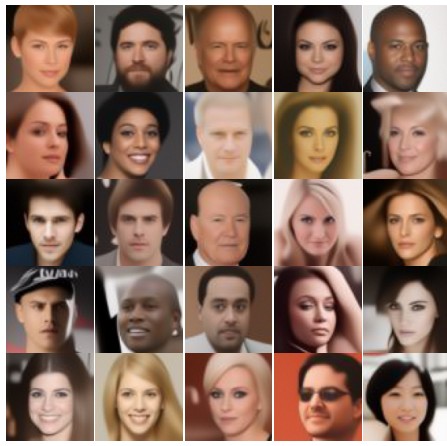

Figure 7: Image samples generated by a GM-SGM trained on CelebA ($64 \times 64$), where we apply $\mathcal{N}(0, 2^2 I)$ instead of Gaussian mixture $\mathcal{N}(0, \{I, 2^2 I\})$ for training. Iterations: 2000.

### A.7   MORE VISUAL EXAMPLES

We first show examples generated by SGMs and GM-SGMs under several settings of iterations number $T$ in Fig. 8-19, then we only show GM-SGMs with even lower iterations on LSUN in Fig. 20-21. Under these low iterations, SGMs only generate high-frequecny noises similar as left of Fig. 8.

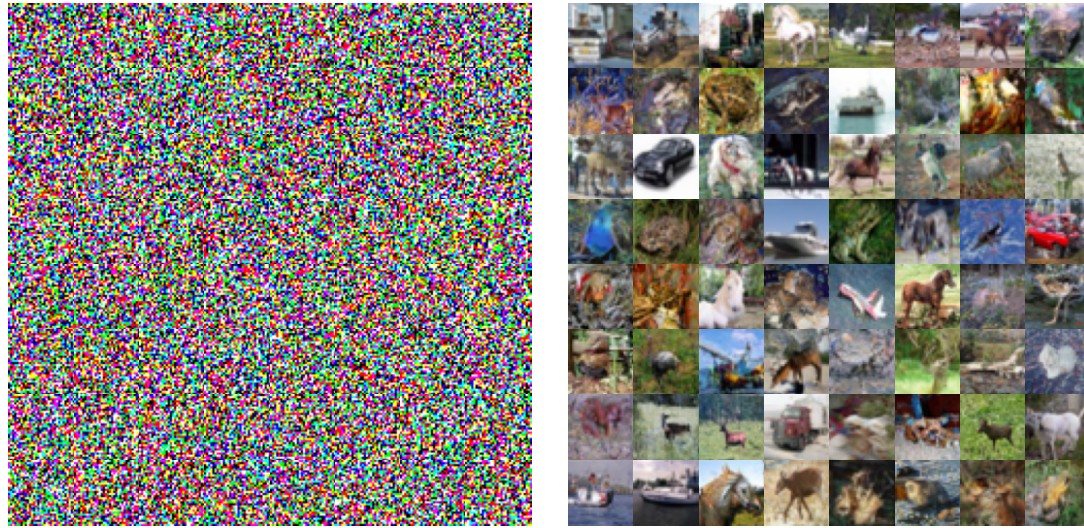

Figure 8: Image samples generated by SGMs (left) and GM-SGMs (right). Dataset: CIFAR-10. Iterations: 50.

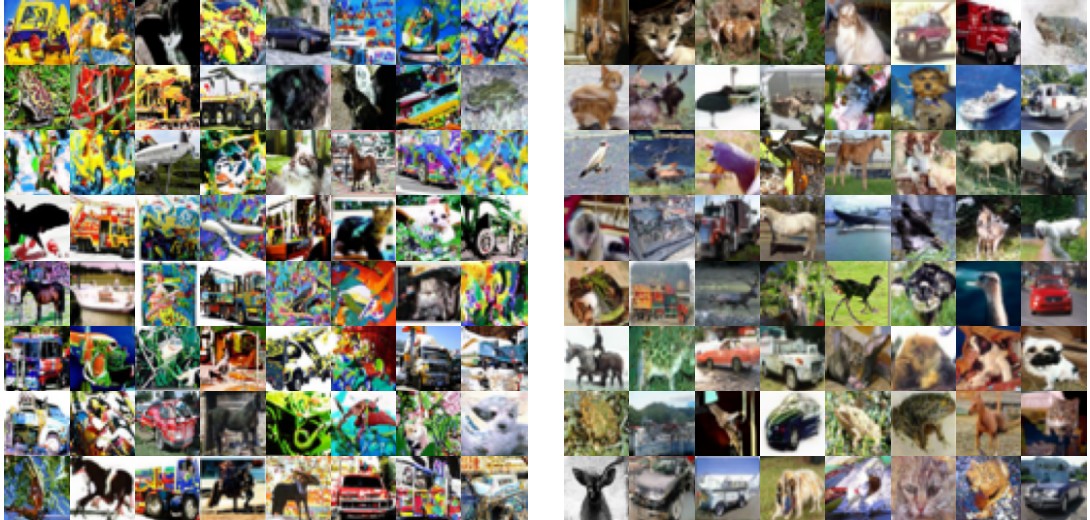

Figure 9: Image samples generated by SGMs (left) and GM-SGMs (right). Dataset: CIFAR-10. Iterations: 200.

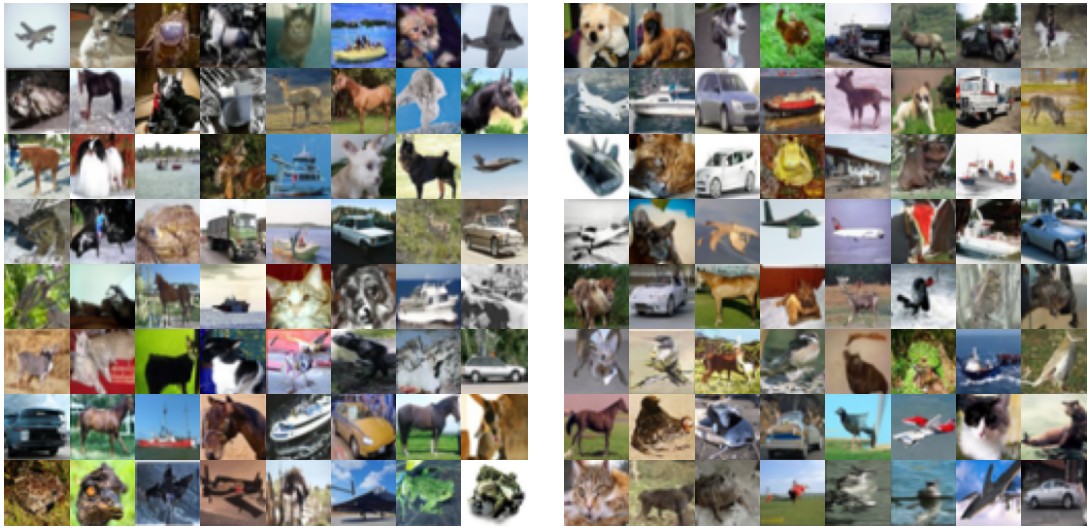

Figure 10: Image samples generated by SGMs (left) and GM-SGMs (right). Dataset: CIFAR-10. Iterations: 500.

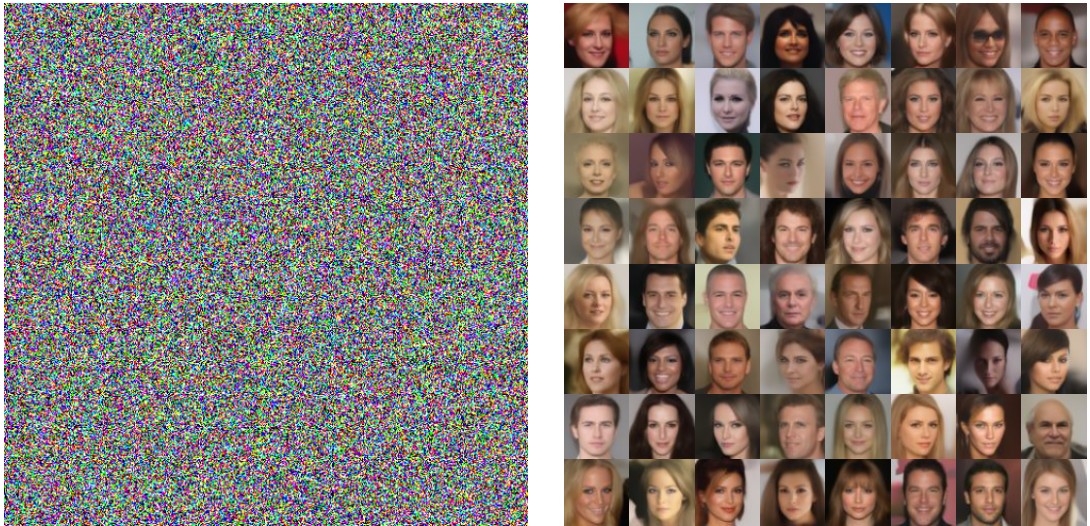

Figure 11: Facial image samples generated by SGMs (left) and GM-SGMs (right). Dataset: CelebA ($64 \times 64$). Iterations: 20.

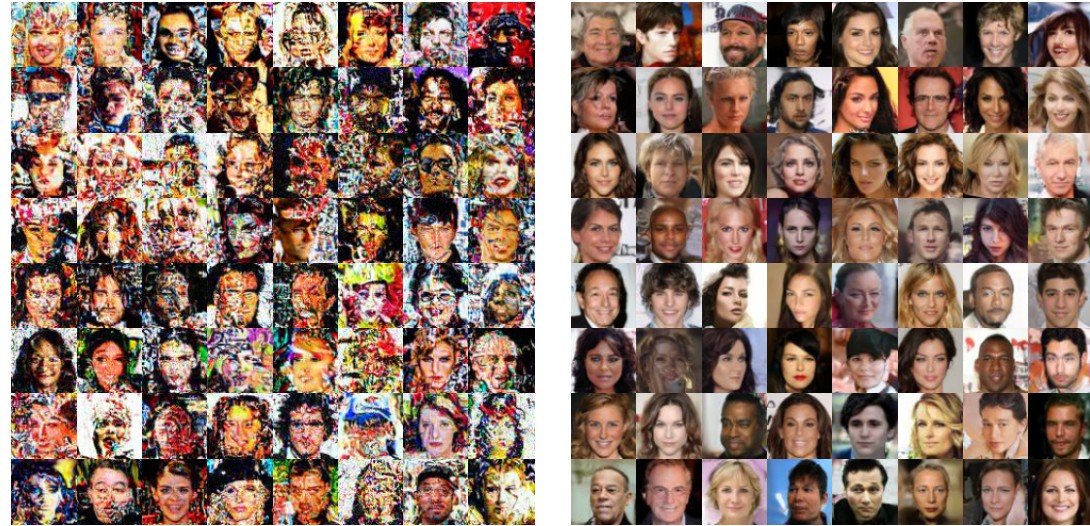

Figure 12: Facial image samples generated by SGMs (left) and GM-SGMs (right). Dataset: CelebA $(64 \times 64)$. Iterations: 200.

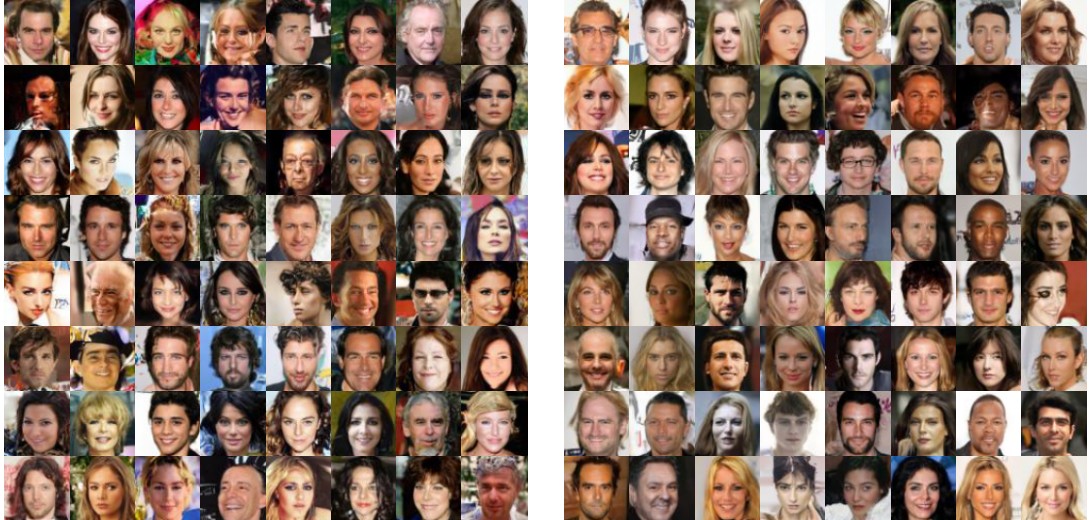

Figure 13: Facial image samples generated by SGMs (left) and GM-SGMs (right). Dataset: CelebA $(64 \times 64)$. Iterations: 400.

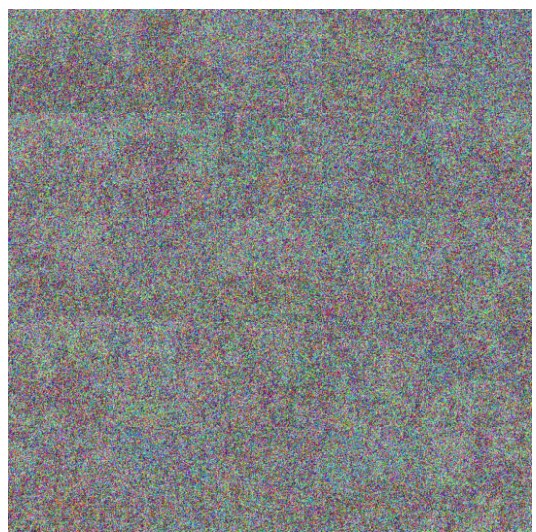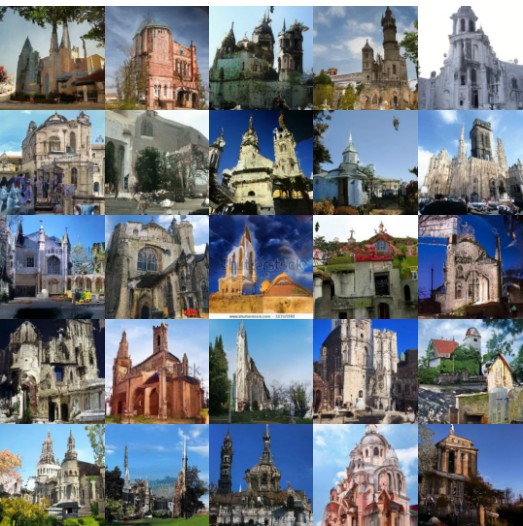

Figure 14: Church image samples generated by SGMs (left) and GM-SGMs (right). Dataset: LSUN (church) ($256 \times 256$). Iterations: 100.

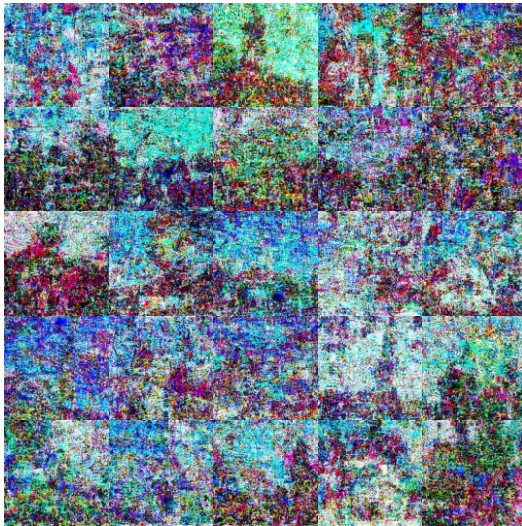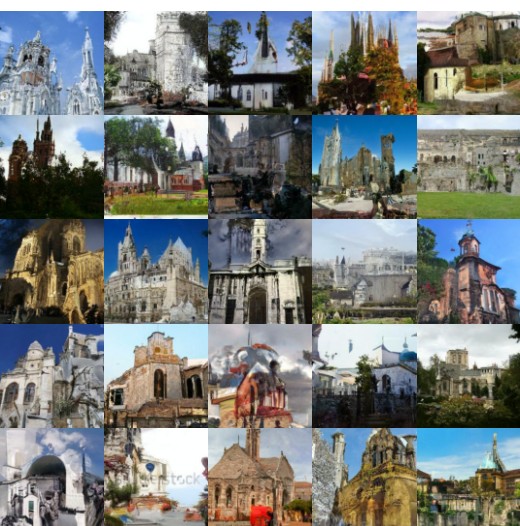

Figure 15: Church image samples generated by SGMs (left) and GM-SGMs (right). Dataset: LSUN (church) ($256 \times 256$). Iterations: 200.

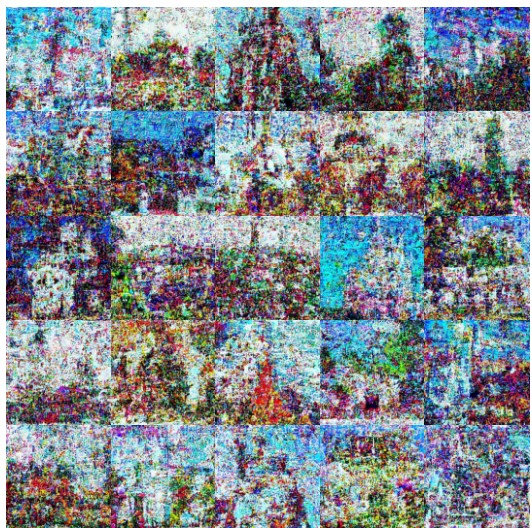 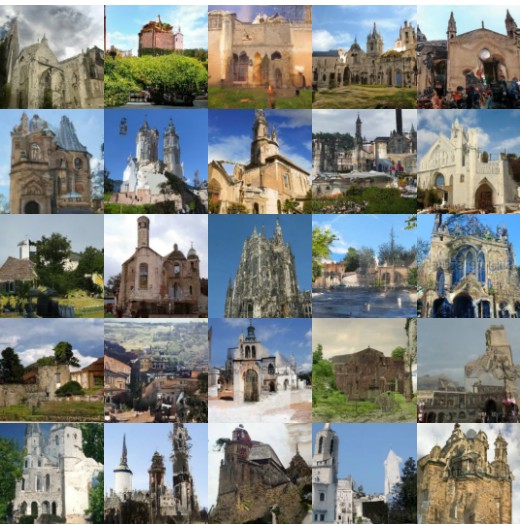

Figure 16: Church image samples generated by SGMs (left) and GM-SGMs (right). Dataset: LSUN (church) ($256 \times 256$). Iterations: 400.

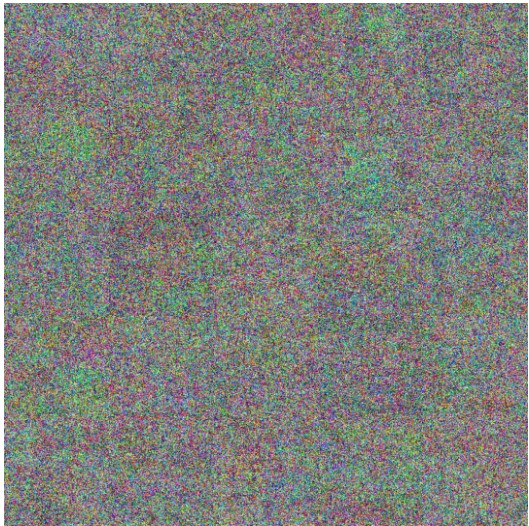 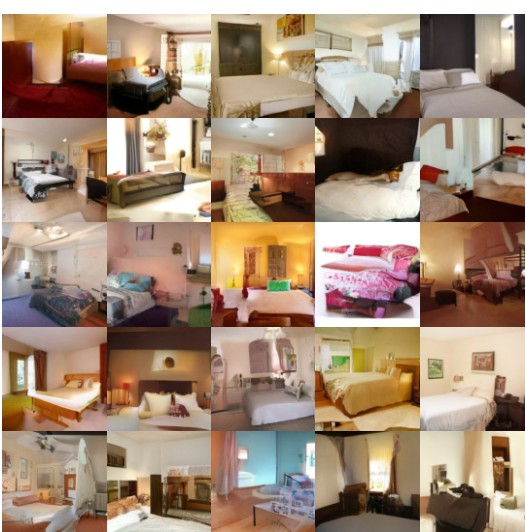

Figure 17: Bedroom image samples generated by SGMs (left) and GM-SGMs (right). Dataset: LSUN (bedroom) ($256 \times 256$). Iterations: 100.

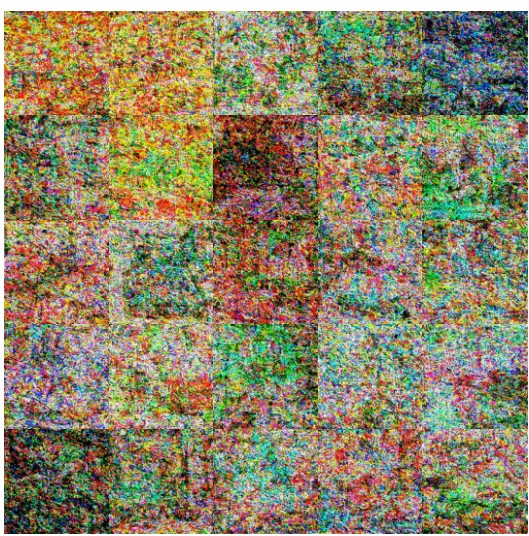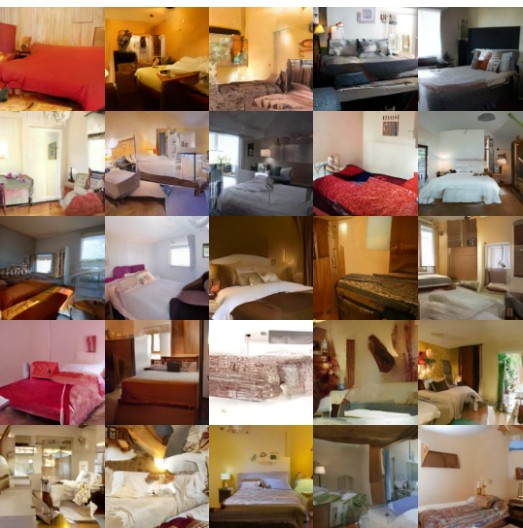

Figure 18: Bedroom image samples generated by SGMs (left) and GM-SGMs (right). Dataset: LSUN (bedroom) ($256 \times 256$). Iterations: 200.

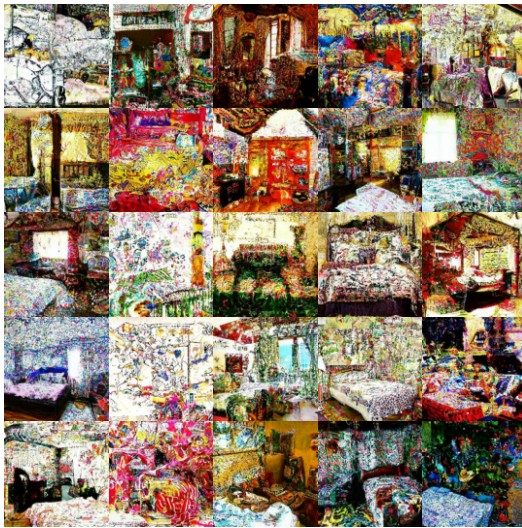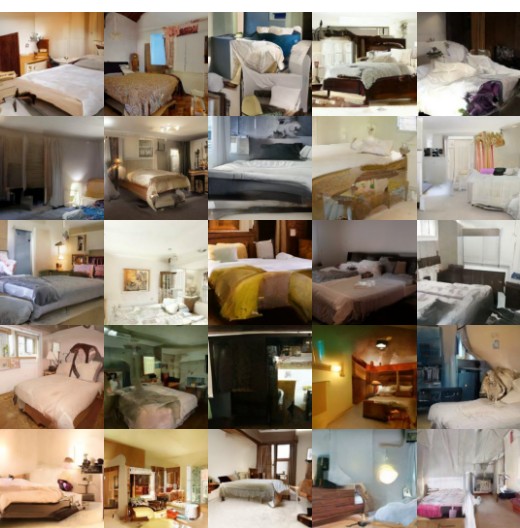

Figure 19: Bedroom image samples generated by SGMs (left) and GM-SGMs (right). Dataset: LSUN (bedroom) ($256 \times 256$). Iterations: 400.

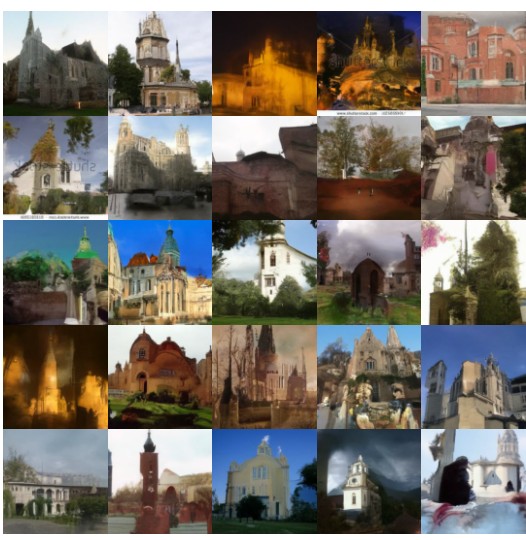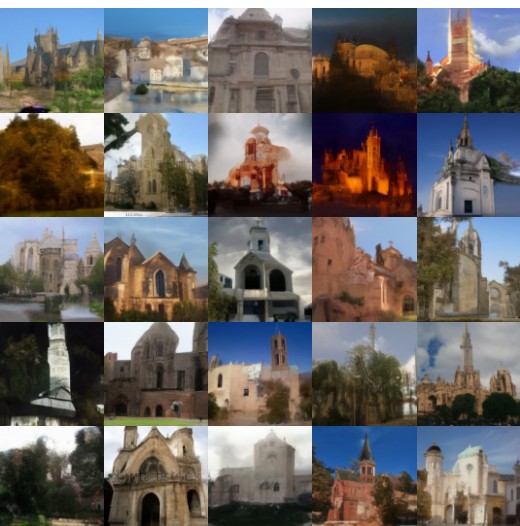

Figure 20: Church image samples generated by GM-SGMs with 50 iterations (left) and 40 iterations (right). Dataset: LSUN (church) ($256 \times 256$). SGMs only produce unrecognizable noises (like left of Fig. 11) below 100 iterations.

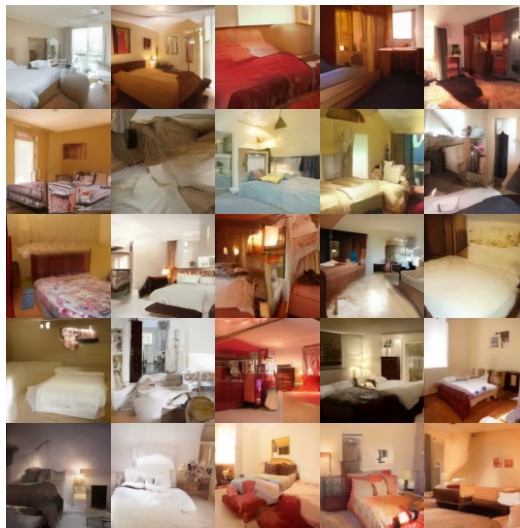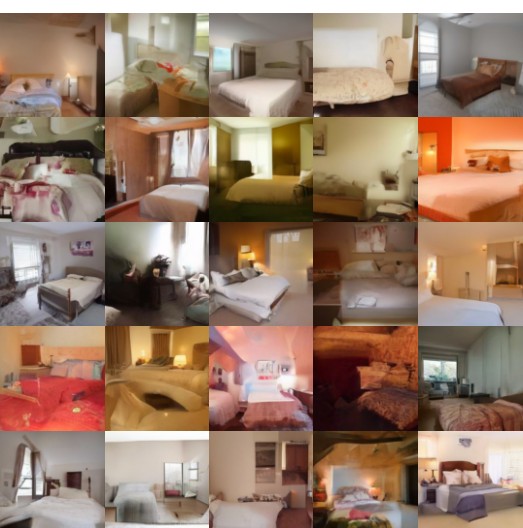

Figure 21: Bedroom image samples generated by GM-SGMs with 50 iterations (left) and 40 iterations (right). Dataset: LSUN (bedroom) ($256 \times 256$). SGMs only produce unrecognizable noises (like left of Fig. 11) below 100 iterations.

