# OpenReview forum: "Approximated Anomalous Diffusion: Gaussian Mixture Score-based Generative Models"
_ICLR.cc/2023/Conference — Submitted to ICLR 2023_

### Official Review · Reviewer_vUWR · 2022-10-20

**Confidence:** 5
**Correctness:** 3
**Technical Novelty And Significance:** 3
**Empirical Novelty And Significance:** 2
**Recommendation:** 5

**Clarity, Quality, Novelty And Reproducibility:**

This work is sufficiently clear, even though the mathematical exposition can be improved and be more rigorous, especially with notation. The quality of this article is good in general, but I really miss a problem statement, motivation and a list of contributions as seen by the authors.

Concerning the novelty, considering alternative distributions to Gaussian to model the noise component has been attempted before (see [2] above). To the best of my knowledge, alpha-stable Levy distributions have not been considered before. So the idea outlined in section 3.1 seems novel to me. Nevertheless, I am not convinced that the proposed approach, although it works in practice, is a good approximation of the Levy diffusion.

I couldn’t check the code for this work, but the Gaussian mixture approximation does not seem to pose particular implementation challenges, and it only affects the training process, whereas sampling is the same as for standard score-based models. As a consequence, I think this work can be reproduced by reading the paper.


**Strength And Weaknesses:**

The strengths of the paper are:
* Improving the efficiency of score-based generative models is an important topic of research, as it is well known that baselines are very costly both in terms of training and sampling

* The experimental results show that the proposed method achieves superior quality of generated images when compared to a vanilla score-based generative model

* This work attempts at porting the body of literature on sampling with alpha-stable Levy noise distributions to diffusion models

The weaknesses of the paper are:
* There is no explicit motivation behind this work! In absence of it, my understanding is that the goal of this paper (based on the emphasis given on the results, and on the concluding remarks) is to accelerate sampling for diffusion models. I think that the authors should clearly state early on in their paper what is the problem they are trying to solve, and why current existing solutions are not sufficient.

* Based on the assumption above, if the goal is sampling acceleration, then the experimental campaign can be improved, by comparing the proposed method to other methods that aim at accelerating sampling, or in general making diffusion models more efficient. For example, several works (for example, [1]) propose adaptive integration schemes to simulate the backward SDE of a score-based diffusion model. I suggest to compare the proposed method to comparable approaches in the literature, instead of using a vanilla baseline which in this case would be “easy to beat”.

[1] https://arxiv.org/abs/2105.14080 “Gotta Go Fast When Generating Data with Score-Based Models”

* The quality of mathematical notation and in general mathematical rigor can be improved. Starting from eq (1), the notation is borrowed from the work on Langevin dynamics from “Bayesian Learning via Stochastic Gradient Langevin Dynamics”, Welling and Teh. But then the subsequent notation comes from early work on score-based generative models, to end up (in sec 3) with the notation and formalism of SDEs. Now, the problem is that step size, noise terms, noise schedule, the diffusion term, etc… use a similar if not the same variable name.  This is very confusing, and should be cleaned up.
I have nothing against presenting the background in reverse, starting from the discretized SDEs, to continuous time SDEs expressions of the diffusion process, but at least the notation should be coherent. Also, some more details (in the background section) could be useful, as for example stating why you focus on variance exploding SDEs, and not variance preserving.

* I understand the definition of an alpha-stable Levy diffusion process is proposed as an original contribution, but I would kindly ask the authors to compare their proposal with similar work that question the use of Gaussian noise only, and proposed to use a different distribution, such as [2]

[2] https://arxiv.org/abs/2110.05948, “Denoising Diffusion Gamma Models”

* Due to the intractability of the reverse Levy process, which requires solving a fractional derivative (for which approximate algorithms exist) and, more importantly, requires access to the data distribution, which is exactly what we are aiming to sample from, the proposed approach is to “approximate” the Levy walk with a Gaussian mixture noise distribution. However, even if the simple 1-D example given in Fig.2 shows some similarities between the trajectories of the proposed approximation and a true Levy walk, in principle the approximation does not guarantee the same properties. There are no large jumps, and the basic principle of “escaping from a local minimum” of a Levy walk cannot be ensured.

* Experiments. In general, experiments are appropriate, as comparing the method to vanilla score-based diffusion is a valid approach to make sure the proposed method works. Nevertheless, the weakness of the empirical evaluation is that it does not compare the proposed method to existing methods from the literature that aim at accelerating sampling. In my view, as there are no clear motivations in the context of score-based generative models, to model the diffusion process as an alpha-stable Levy diffusion, the main take home message I get from the experiments is that it produces better samples faster than the vanilla model. Then, it makes sense to ask how the proposed method fares when compared to, say, better SDE integrators.


**Summary Of The Paper:**

The work presented in this paper stems from the observation that sampling from a target distribution can benefit from heavy-tailed Levy dynamics, whereby the stochastic process is allowed to implement large jumps, thus avoiding local minima and allowing the process to explore the distribution space more efficiently.

Thus, the idea is to extend score-based generative models, a la Song et al.21b, such that the SDE dynamics describing the forward and backward diffusion processes can benefit from a Levy-like behavior, and allow large jumps in the corruption and denoising steps.

However, the inversion of an alpha-stable Levy process is intractable, and the authors suggest approximating the Levy process by inducing a Gaussian mixture model on the Brownian motion term of diffusion of the vanilla score-based SDE model. The hope is that this approximation faithfully reproduces the properties of an alpha-stable Levy distribution. It is important to note that while training the proposed model requires the Gaussian mixture of noise variances, samping from it is instead done using the vanilla approach of simulating the reverse SDE with a single, time-varying Gaussian noise distribution.

A series of experiments complement the methodological contribution of this work, whereby the authors compare their model to the vanilla score-based diffusion model, using three popular datasets for the task. In summary, the results indicate that the proposed method outperforms generative quality (in terms of FID score) of vanilla score-based models, at a fraction of the number of sampling steps.


**Summary Of The Review:**

This work presents an interesting perspective, although it lacks a clear motivation. The attempt to solve the intractability of the proposed alpha-stable Levy diffusion process is not convincing in my opinion. The proposed approach achieves good empirical results, when compared to a simple baseline, but it has not been compared to alternative methods to accelerate sampling. There are many other weaknesses in the submitted paper, stemming from lack of rigor in the mathematical description of the model, as well as in the proposed approximation of the Levy diffusion process.


=============================

Post rebuttal comment

=============================

I've read the rebuttal, all reviews and their comments. I am willing to slightly raise my score, but I am not convinced this submission deserves to be accepted at this time. I hope the Authors will find comments useful to further improve their work.

---

> ### Author Response · Authors · 2022-11-06
> **Response to reviewer vUWR**
>
> Thank you for your detailed reviews of our work.
> >*Q1: There is no explicit motivation behind this work! In absence of it, my understanding is that the goal of this paper.*
>
> Apologies for this biased understanding.
> While faster than SGMs, acceleration is not the core of our GM-SGMs formulation, but a resulting good characteristic.
> As clearly stated in Introduction and Sec.3, our motivation is sourced from the anomalous diffusion of biological neural systems, in particular its advantages over the Langevin dynamics of existing SGMs.
> Under this insight, we investigate a more general class of SGMs not limited to the standard Langevin dynamics.
>
> Our proposed GM-SGMs, as analyzed in 4.2,
> underpin several fundamental aspects:
> (1) Training SGMs with more robustness;
> (2) Demonstrating how ensemble learning can be applied to the SGMs;
> (3) Manifesting the possibility to implement a sampling process of multiple paths ($2^T$ paths), as validated by the result in the first column of Fig.4.
>
> These are much more than SGM acceleration.
>
>
> >*Q2: Based on the assumption above, if the goal is sampling acceleration, I suggest to compare the proposed method to comparable approaches in the literature (for example, [1]), instead of using a vanilla baseline which in this case would be “easy to beat”.*
>
> As responded above, model acceleration is not essential with our GM-SGMs. However, we still conduct a comparative experiment with [1] (a preprint, not formally published). For a comprehensive comparison, we consider both the variance preserving (VP) and variance exploding (VE) versions of the proposed method in [1] (denoted as adaptive SDE solver (VE)).
> This test is conducted on CIFAR-10 with 50 sampling iterations.
> The results in the table below show that our GM-SGM is significantly superior over [1].
>
> | Method | adaptive SDE solver (VP) | adaptive SDE solver (VE) | SGM  | GM-SGM (ours) |
> | --- | --- | --- | --- | --- |
> | iterations | 49 | 50 | 50 | 50 |
> | FID | 82.42 | 307.32 | 456.49 | 29.41 |
>
> We will add this test.
>
>
> >*Q3:The quality of mathematical notation and in general mathematical rigor can be improved. Also, some more details (in the background section) could be useful, as for example stating why you focus on variance exploding SDEs, and not variance preserving.*
>
> Thank you for these suggestions.
> For notations, we will refine them as suggested.
> For the reason of focusing on variance exploding (VE) SDEs instead of variance preserving (VP)is that,
> the VE is found similar to the Langevin Dynamics, hence the techniques from the anomalous diffusion can be more naturally extended. In contrast, the sampling process of VP has no corresponding anomalous diffusion process, since it is not a Langevin Dynamics.
> Nevertheless, we believe that our idea can be generally applied to VP; This might lead to a different understanding about how the mixture Gaussian noises work. We will leave this to further study.
>
>
> >*Q4:I would kindly ask the authors to compare their proposal with similar work that question the use of Gaussian noise only, and proposed to use a different distribution, such as [2]*
>
> Thanks. As suggested, we compare DDGMs [2] (a preprint, not formally published) with our GM-SGMs.
> For a quantitative comparison on CelebA (64x64) as shown in the table below, our GM-SGMs are clearly superior over DDPMs, DDPM-based DDGM[2], and comparable to DDIM and DDIM-based DDGM[2] despite that their base model DDPM and DDIM already have better low-iteration performance than SGMs.
>
> | Method | DDPM | DDIM | DDGM (DDPM-based) | DDIM (DDIM-based) | SGM | GM-SGM (ours) |
> | --- | --- | --- | --- | --- | --- | --- |
> | 20 | 183.83 | 13.73 | 28.24 | 6.83 | 439.32 | 27.81 |
> | 100 | 45.2 | 6.53 | 14.22 | 3.17 | 435.59 | 5.99 |
>
> In terms of computational efficiency, compared to DDGMs our GM-SGMs are also better due to adopting the vanilla sampling process (i.e., simply replacing the checkpoint model during inference).
> In contrast, DDGMs induce much more extra complexity.
>
> In terms of formulation, as discussed in the response to Q1 above, our models are inspired by the probabilistic representation of biological neural systems ( Sec. 1 and Sec. 3.1),
> along with several key characteristics (Sec. 4.2).
> These are all lacking in the use of Gamma distribution with DDGMs.
>
> >*Q5:There are no large jumps, and the basic principle of “escaping from a local minimum” of a Levy walk cannot be ensured.*
>
> As illustrated in Fig. 2, large jumps must be in place with our formulation (middle), otherwise, the trajectories would not spread this widely, much more than Brownian motion (right) with SGMs.
> Formally, we train our GM-SGMs to consider both large jumps and small roaming by inputting noises from $\mathcal{N}(0,\beta^2 I)$  and $\mathcal{N}(0,I)$ for fitting the score function (Eq.9).
> The former type of noises can be regarded as the outliers of $\mathcal{N}(0,I)$, making the denoising of the GM-SGM  more likely take large steps to escape from a local minimum.

---

> ### Author Response · Authors · 2022-11-20
> **Request for feedback on the rebuttal**
>
> Dear Reviewer vUWR,
> We appreciate all the reviewing time and effort.
> With our best appreciation, we have made the response in detail to each individual comment.
> While we consider this could have addressed all the concerns raised hopefully, it is most critical that the reviewer can kindly read our response and tell us how the issues have been addressed and if any concerns are left to be addressed.
> We would take all the comments/suggestions as carefully as possible and address them with our best efforts.
> Many thanks for every effort the reviewer made and will make on our work.
>
> Best wishes,
> Authors

---

> > ### Comment · Reviewer_vUWR · 2022-11-29
> > **Thank you for your comments**
> >
> > Dear Authors,
> > I've read your rebuttal to my review, the other reviews and corresponding discussions.
> >
> >
> > Thank you for the clarification about the overall objective of your work, which was not clear to me. It is indeed an intriguing exercise to explore the possibility of using Levy dynamics instead of Langevin dynamics, with the hope to improve (among other aspects) the robustness of the diffusion process.
> >
> > On this point -- and I share the same view of another reviewer -- I am still not convinced about the idea of robustness to outliers in the noise values. I have read your points, and I can see in the experiments that it seems to lead to lower FID scores (which is what we seek for, ultimately), but in my experience with diffusion models the noise is not sampled uniformly at random anymore, opting for "importance sampling": noise levels are more frequently sampled closer to the data distribution than to the noise/"prior" distribution. This has the effect of decreasing the FID score. Does this mean that the current practice makes diffusion models even less robust? Overall, the discussion on robustness is still not rock solid, in my humble opinion.
> >
> > Finally, I understand that the proposed GMM to approximate a Levy walk is just an approximation, which is fair.
> >
> > Now, I am willing to slightly increase the score of this submission, but I'm still not 100% happy about it. It might be "old-style" or simplistic, but I really miss a clear narrative for this work, such as: "Vanilla score-based models have this huge problem, as shown in our experiments. These problems are due to the Langevin dynamics, which are not robust in dealing with noise outliers. We propose to replace Langevin with Levy because it has been shown [cite, cite, cite] that these stochastic processes are more robust to the one main culprit for bad performance of Langevin processes. However, Levy processes are not analytically tractable, so we define a heuristic that resembles to Levy processes, and that inherits their robustness. This translates into 1) higher generative quality, 2) shorter sampling times, at a manageable cost." Instead, the narrative in the revised submission is still vague, and while there has been an effort to point at problems with vanilla diffusion through the lenses of generative performance, I am not fully grasping the problem we are trying to solve here.
> >
> > Let me conclude by thanking the authors for their work, and their rebuttal.

---

> > > ### Author Response · Authors · 2022-12-01
> > > **Response to reviewer vUWR**
> > >
> > > We really appreciate the reviewer vUWR for giving further comments and useful feedback with great effort.
> > >
> > > We stress again that overall we aim to improve the generation ability of the SGMs by exploring the merits of the Levy dynamics.
> > > Our motivation is from neural computation, as stated from the very beginning (e.g., Abstract).
> > > As explained in Sec.3.2, the exact formulation of Levy dynamics is computationally intractable and hence not directly computable.
> > > This is a big challenge that has not been explored to the best of our knowledge;
> > > We appreciate related pointers and suggestions if any.
> > > Under this context, as the first ever attempt we propose an efficient tractable solution with Gaussian mixture for implementing the key capabilities (i.e., large jump and small roaming) of the Levy dynamics, which we consider is a significant contribution in design.
> > > We hope the reviewer could re-evaluate our work under the above circumstances.
> > >
> > > To minimize the misunderstanding, it is worthwhile to point out that there is not direct relationship between noise robustness
> > > and Levy dynamics.
> > > Instead, noise robustness is one of the resulting characteristics (including the ensemble learning and multiple path sampling).
> > > This analysis aims to help understand why our GM-SGMs are superior over existing SGMs in generating samples with much higher quality
> > > (as explained in Sec 4.2).
> > > Please do not get confused with our motivation as mentioned above.
> > >
> > > Regarding the question about robustness (
> > > *Do the current practices make diffusion models even less robust, where noise levels are more frequently sampled closer to the data distribution than to the noise/prior distribution?*
> > > ).
> > > Some existing improvements have been made for improving the model's robustness in varying manners.
> > > For example, in order to improve DDPMs without model retraining, [1] alters the noise levels and step schedules during sampling so that the sample point gets closer to the manifold of the trained forward process. Their hypothesis is that if a sample point is far away from the trained manifold, the score function would become less reliable, particularly in case of low iterations (please see Sec.3.2 in [1]).
> > > Instead of altering the sampling process, we directly train a new diffusion model using amplified noises, so that the network can fit the score function under a much wider range of noises (even far away from the manifold of the forward process). This is the reason why we consider GM-SGM is more robust.
> > > Note, robustness is a fundamental yet under-studied facet of SGMs.
> > > We also note that, when compared to previous sampling acceleration methods, our model can automatically adjust the step schedule during sampling, without the need for manually refining the noise schedules [2], or using high-order SDE (ODE) solvers for altering the sampling process [3].
> > >
> > > [1] Liu, L., Ren, Y., Lin, Z., & Zhao, Z. (2022). Pseudo numerical methods for diffusion models on manifolds. arXiv preprint arXiv:2202.09778.
> > >
> > > [2] Bao, F., Li, C., Zhu, J., & Zhang, B. (2022). Analytic-dpm: an analytic estimate of the optimal reverse variance in diffusion probabilistic models. arXiv preprint arXiv:2201.06503.
> > >
> > > [3] Lu, C., Zhou, Y., Bao, F., Chen, J., Li, C., & Zhu, J. (2022). DPM-Solver: A Fast ODE Solver for Diffusion Probabilistic Model Sampling in Around 10 Steps. arXiv preprint arXiv:2206.00927.

---

> > > ### Author Response · Authors · 2022-12-06
> > > **Response to reviewer vUWR**
> > >
> > > Dear Reviewer vUWR,
> > >
> > > We appreciate all the reviewing time and effort.
> > >
> > > With our best appreciation, we have responded to the comments about our motivation and clafiried the meaning of noise robustness. We hope this could address all the concerns.
> > >
> > > It is thankful that the reviewer could further review and let us know if any issues/concerns for us to clarify and solve.
> > >
> > > Best wishes,
> > > Authors

---

### Official Review · Reviewer_HQ5G · 2022-10-21

**Confidence:** 4
**Correctness:** 3
**Technical Novelty And Significance:** 2
**Empirical Novelty And Significance:** 3
**Recommendation:** 5

**Clarity, Quality, Novelty And Reproducibility:**

Clarity: i) The results of SGM in Tables 1 and 2 seem too strange to me, there seems to be a cut-off phenomenon for SGM according to the authors. However, I am not convinced why this happens; ii) The connection to the probabilistic graphical model is not detailed enough.



Novelty: The idea of exploring heavy tail distributions for exploration is interesting; Gaussian mixture approximation is acceptable, but the implemented method is not clever enough.

Quality: The method itself is not comprehensive enough; e.g. at least the authors should clarify what is the **optimal mixture formulation given a Levy process of a certain degree.**

Reproducibility: methodology seems simple enough to implement; didn't check the code. I am not convinced why there is a cut-off phenomenon for SGM in table 1 and 2.

Minor: define GM before using it in abstract.

**Strength And Weaknesses:**



Pros: the insight of exploring the mixture of Gaussian distributions to approximate heavy tail distributions is quite appealing. I like this idea and the local trap problem in generative tasks is always a critical concern. There are various popular methodologies to tackle this issue, but no one has studied the important Levy dynamics before. As such, it is worthwhile to study related algorithms and how such an idea performs in applications such as generative models.

Cons:

1. My biggest concern is that the mixture formulation is not general enough; why it has to be 50% and 50%, just because it is easier to implement (n mod 2 ==0)? Can we achieve it in a better way, say probabilistically to achieve more flexible settings?

2. the transition from Levy process to Gaussian mixture is a bit fast. For certain types of Levy process, what is the optimal Gaussian mixture under some formulations to approximate it?

3. In section 4.2, the authors said the mixture update is more robust, but not enough proof can support that.

4. The mixture formulation is first presented in section 3.2 and then becomes continuous in section 4.1, but then goes to discrete again, which looks confusing.

**Summary Of The Paper:**

This paper proposed a class of score-based generative models motivated by Levy dynamics to produce large jumps and small roaming to facilitate exploration. Since the exact numerical simulation of Levy dynamics is quite challenging and intractable, the authors proposed a vanilla mixture formulation to approximate Levy dynamics; the authors relate such an update with the probabilistic graphical model for illustration purposes. The authors also empirically verified that the exploration property has appealing properties in generative models.

**Summary Of The Review:**

The idea of adopting Levy process to facilitate exploration is interesting; the Gaussian mixture approximation is acceptable; however, the method itself is not good enough in my opinion. As such, I tend to be conservative in my ratings.

---

> ### Author Response · Authors · 2022-11-06
> **Response to reviewer HQ5G**
>
> Thank you for taking the time to read our paper and provide these detailed reviews.
> >*Q1: My biggest concern is that the mixture formulation is not general enough; why it has to be 50\% and 50\%, just because it is easier to implement (n mod 2 ==0)? Can we achieve it in a better way, say probabilistically to achieve more flexible settings?*
>
> Great comment. Yes, our current implementation is simple to implement and not necessarily optimal. In general, it is possible to find even better splits with a grid search.
> We will add an experiment in the final version.
>
> >*Q2: The transition from Levy process to Gaussian mixture is a bit fast. For certain types of Levy process, what is the optimal Gaussian mixture under some formulations to approximate it?*
>
> Apologies for some misunderstanding here.
> To be more precise,
> our GM-SGMs approximate the anomalous diffusion in the sense of functional similarity, instead of approximating the Levy distribution in the anomalous diffusion.
> Concretely, we aim to use the Gaussian mixture to enable the sampling of SGMs to implement large jump and small roaming as the anomalous diffusion, rather than rigidly approximating the Levy process through Gaussian mixture.
> As mentioned in Section 3.2, the Riesz fractional derivative in Eq.(7) is a hard obstacle to simulate an anomalous diffusion for sampling, while sampling from Levy distribution is just one of the outstanding difficulties
> As a result, for high-dimensional data generation such as image generation, SGMs with anomalous diffusion is infeasible even assume we have managed to approximate the Levy distribution.
> To bypass this, we adopt a more scalable approach to capturing its functional merits (large jump and small roaming), but not struggle to simulate a genuine anomalous diffusion process.
>
> >*Q3: In section 4.2, the authors said the mixture update is more robust, but not enough proof can support that.*
>
> Thanks. We stress again that the robustness comes from the fact that our GM-SGM is trained to fit the score function, where the input noises can be standard Gaussian distribution (original training objective) or have higher variance (the noises that are amplified by $\beta$) than the standard Gaussian distribution (see Eq.11).
> These amplified noises could be considered as the outliers of the standard Gaussian distribution since they appear at much smaller probabilities.
> As a result, GM-SGMs are rendered more robust than vanilla SGMs.
> We have empirically validated this robustness:
> This has been verified in the experiment part.
> (1) We first show that GM-SGMs have a much higher mean of the norms of predicted noises compared to the vanilla SGMs (Fig. 4), suggesting that GM-SGMs take into consideration more outliers than the vanilla SGMs;
> (2) We show that GM-SGMs yield superior FID scores over SGMs (Tables 1,2), suggesting that such outliers are necessary to be considered in denoising.
>
> >*Q4:  The mixture formulation is first presented in section 3.2 and then becomes continuous in section 4.1, but then goes to discrete again, which looks confusing.*
>
> Thank you for pointing this out. This is because, in the context of continuous formulation, we focus on introducing our motivation from both theoretical and biological sources; And in the context of discrete formulation, we focus on specifying our method with more incline towards implementation.
> We will further clarify this.

---

> ### Author Response · Authors · 2022-12-01
> **Response to reviewer HQ5G (II)**
>
> Dear Reviewer HQ5G,
>
> We appreciate all the reviewing time and effort. With our best appreciation, we have made the response in detail to each individual comment. While we consider this could have addressed all the concerns raised hopefully, it is most critical that the reviewer can kindly read our response and tell us how the issues have been addressed and if any concerns are left to be addressed. We would take all the comments/suggestions as carefully as possible and address them with our best efforts. Many thanks for every effort the reviewer made and will make on our work.
>
> Best wishes, Authors

---

### Official Review · Reviewer_9CCy · 2022-10-23

**Confidence:** 2
**Correctness:** 3
**Technical Novelty And Significance:** 2
**Empirical Novelty And Significance:** 2
**Recommendation:** 3

**Clarity, Quality, Novelty And Reproducibility:**

Clarity: The paper is easy to follow.

Quality: I have never done any research in this area, but I believe the quality and novelty of this paper is high.

Reproducibility: Unknown.

**Strength And Weaknesses:**

Strength: replacing Gaussian with Gaussian mixtures seems interesting, but I am not sure since I have never done any research in this area.


Weakness:

1. Why can we use Gaussian mixtures to approximate the Levy distribution? How accurate the approximation is? I wonder if there is any theoretical guarantee. Intuitive, Gaussian mixtures are still lighted-tailed. I would like to see some rigorous analysis of these.


2.  It seems to me that by doing parallel tempering of Langevin dynamics and related techniques, we can also achieve both small roaming mode and large jump mode.

3. The stationary distribution of Langevin dynamics is known. However, when replacing the Gaussian noise with the Levy distribution or even Gaussian mixtures, the stationary distribution will be changed completely, and actually, we do not know what is the stationary distribution anymore.


Question to the author:

After reading the paper, I am still not sure how to perform the reverse procedure when the Gaussian mixture is applied in the forward procedure. Can authors comment on this?



**Summary Of The Paper:**

Based on heavy-tailed Levy dynamics can produce both large jumps and small roaming to explore the sampling space, resulting in better sampling results than Langevin dynamics with a lacking of large jumps, the authors explore a new class of score-based generative models (SGMs) with sampling based on the Levy dynamics. However, the exact numerical simulation of the Levy dynamics is significantly more challenging and intractable. The authors propose an approximation solution by leveraging Gaussian mixture noises during training to achieve the desired large jumps and small roaming properties.

**Summary Of The Review:**

It seems to me that this work simply replaces Gaussian noise with Gaussian mixtures in SGMs. However, there are some questions that are not clear to me. 1) Why Gaussian mixtures can approximate heavy-tail distributions? This seems invalid to me. 2) Do we really need to replace Gaussian noise in the Langevin dynamics with others to improve sampling? As far as I am aware, the mainstream technique is to improve sampling techniques, e.g., adaptive bias force, parallel tempering, etc.

---

> ### Author Response · Authors · 2022-11-06
> **Response to reviewer 9CCy**
>
> Thank you for taking the time to provide these detailed reviews.
> > *Q1: Why can we use Gaussian mixtures to approximate the Levy distribution? How accurate the approximation is? I wonder if there is any theoretical guarantee. Intuitive, Gaussian mixtures are still lighted-tailed. I would like to see some rigorous analysis of these.*
>
> Apologies for some misunderstanding here. To be more precise, our GM-SGMs approximate the anomalous diffusion in the sense of functional similarity, instead of approximating the Levy distribution in the anomalous diffusion.
> Concretely, we aim to use the Gaussian mixture to enable the sampling of SGMs to implement large jump and small roaming as anomalous diffusion, rather than rigidly approximating the Levy distribution through the Gaussian mixture.
> As mentioned in Section 3.2, the Riesz fractional derivative in Eq.(7) is a hard obstacle to simulating an anomalous diffusion for sampling, while sampling from Levy distribution is just one of the outstanding difficulties.
> As a result, for high-dimensional data generation such as image generation, SGMs with anomalous diffusion is infeasible even assuming we have managed to approximate the Levy distribution.
> To bypass this, we adopt a more scalable approach to capturing its functional merits (large jump and small roaming), but not struggle to simulate a genuine anomalous diffusion process.
>
> > *Q2: It seems to me that by doing parallel tempering of Langevin dynamics and related techniques, we can also achieve both small roaming mode and large jump mode.*
>
> Agreed. Indeed, the vanilla SGMs already use  similar techniques such as simulated annealing for better exploring the sampling space. However, this is still limited. Our GM-SGMs can further strengthen this ability, yielding significant extra improvement as shown in our experiments.
>
> > *Q3: The stationary distribution of Langevin dynamics is known. However, when replacing the Gaussian noise with the Levy distribution or even Gaussian mixtures, the stationary distribution will be changed completely, and actually, we do not know what is the stationary distribution anymore.*
>
> Thanks. With our GM-SGMs, the original stationary distribution can be preserved.
> This is because we still use the original Gaussian noise during sampling instead of the mixture Gaussian.
> We only use the mixture Gaussian while training the GM-SGMs, but keep the sampling process the same as the vanilla SGMs.
> We have already provided a theoretical explanation in Sec.4.1.
> Due to this issue, using a mixture Gaussian during sampling could degrade the performance (still outperforming vanilla SGMs, see Sec.5.4).
>
> Please note that we have already analyzed the convergence of GM-SGMs in Sec. 4. It is found that both GM-SGMs and SGMs have the same aim of inference (the noises added to the sample point) for denoising during sampling.
> The key difference is that GM-SGMs implement the probabilistic graphical model of an empirical Bayes procedure, versus that of a Maximum a posteriori (MAP)
> by SGMs.
> By considering the possibility of both small roaming and large jumps during sampling, our GM-SGMs are more robust to the outlier of Gaussian noises, resulting in a stronger sampling ability.

---

> > ### Comment · Reviewer_9CCy · 2022-11-20
> > **Thank you for your response**
> >
> > I acknowledge that I have read the rebuttal. Overall, I still feel the work is not significant, and some of my questions are not addressed. I am new to this area and have only read a few papers on diffusion models. I maintain my original rating on this paper.

---

> > > ### Author Response · Authors · 2022-11-20
> > > **Response to Reviewer 9CCy**
> > >
> > > Many thanks for this feedback and honest opinion on the difficulty of evaluating our work.
> > >
> > > Could you please specify the quetions we have not addressed well if possible. If they are about backbround knowledge, please also feel free to let us know. We are more than happy to further clarify.

---

> ### Author Response · Authors · 2022-11-20
> **Request for feedback on the rebuttal**
>
> Dear Reviewer 9CCy,
> We appreciate all the reviewing time and effort. With our best appreciation, we have made the response in detail to each individual comment.
> While we consider this could have addressed all the concerns raised hopefully, it is most critical that the reviewer can kindly read our response and tell us how the issues have been addressed and if any concerns are left to be addressed. We would take all the comments/suggestions as carefully as possible and address them with our best efforts. Many thanks for every effort the reviewer made and will make on our work.
>
> Best wishes,
> Authors

---

### Official Review · Reviewer_ULSW · 2022-10-24

**Confidence:** 3
**Correctness:** 3
**Technical Novelty And Significance:** 3
**Empirical Novelty And Significance:** 3
**Recommendation:** 8

**Clarity, Quality, Novelty And Reproducibility:**

The paper has a clear description of the idea and the issues involved. It is a new approach to introduce Levy dynamics to the score-based generative model. The reproducibility is also fine.

**Strength And Weaknesses:**

Strength:
1. Using Levy dynamics allows sample points to generate large jumps to pass through low-probability regions, making it less likely that they will remain at local minima.

2. Because of the high density near the zero center in Lévy dynamics, there is an expectation that local regions can be searched efficiently.

3. Experimental results show that a mixture of sampling from Gaussian distributions, which are computationally less expensive to approximate, instead of sampling from a Lévy distribution, still captures the desired behavior.

4. GM-SGM can automatically select large jumps or small roaming when sampling.


Concerns:
1. How much computation time/cost is needed for inference at each step t compared to SGM?

2. If GM-SGM can achieve high performance with a small number of iterations, can it achieve a better score with more iterations, or is the limit the current FID score (around 3.5)?

**Summary Of The Paper:**

The authors argue that the Lévy dynamics sampling method is preferable to sampling based on Langevin dynamics in score-based generative models (SGM). The reason is based on the hypothesis that since Lévy dynamics has a heavy-tailed distribution, it can produce both large jumps and small roaming in the search for the sampling space, and thus can provide better sampling results than Langevin dynamics, which is not expected to produce large jumps. On the other hand, since exact numerical simulation of Lévy dynamics is difficult, the authors propose to use Gaussian mixture noises as an approximate method. Specifically, the authors propose Gaussian mixture SGM (GM-SGM) as a new type of SGM that learns to denoise Gaussian mixture noises.

**Summary Of The Review:**

The use of Lévy dynamics to create large jumps is a major contribution for achieving highly accurate image generation with a small number of steps. A discussion of the computation time would be helpful.

---

> ### Author Response · Authors · 2022-11-06
> **Response to reviewer ULSW**
>
> Thank you for the time you have allocated in reading our paper and providing the detailed review.
> > *Q1:How much computation time/cost is needed for inference at each step t compared to SGM?*
>
> As stated in Sec.3.4, our GM-SGMs use the same sampling process as the vanilla SGMs (NCSN++), so giving no extra cost per step.
>
> > *Q2: If GM-SGM can achieve high performance with a small number of iterations, can it achieve a better score with more iterations, or is the limit the current FID score (around 3.5)?*
>
> Great comment. Following this suggestion, we find out that when sampling a specific noise per sample, our GM-SGM can bring FID down to 2.25, vs. 2.50 by SGM. We will add this experiment.

---

> > ### Comment · Reviewer_ULSW · 2022-11-30
> > **Thanks for your response comments**
> >
> > Dear Authors,
> >
> > 　Thanks for your rebuttal. I carefully read not only your responses but also the discussions with other reviewers. After reading the correspondence between the author and the other reviewers, I was able to grasp once again the possibility that the proposed method may realize a new way of sampling of SGMs to implement large jump and small roaming. I recognized the motivation. As other reviewers pointed out, I suppose there is still room for debate regarding robustness and theoretical superiority over vanilla SGMs. Meanwhile, I would like to evaluate the experimental demonstration of the point that even an approximation method of the Lévy distribution based on heuristics can provide comparable or superior results than SOTA, as well as the analysis and discussion of the causes for obtaining such results. For these reasons, I would like to maintain my positive evaluation of this paper.

---

### Author Response · Authors · 2022-11-13
**General Responses to All Reviewers**

We sincerely thank all reviewers for their constructive suggestions and valuable comments.
We have carefully responded to all the suggestions/comments.
Below we recap the key contributions of this work for facilitating the following review process.

+ As mentioned in Sec 3.3-3.4,
our proposed GM-SGMs do not change the original sampling during inference and make small modifications during training with marginal extra complexity.
Despite this simplicity, our model can yield significant generative improvement.

+ As introduced in  Sec. 1 and Sec. 3, our motivation comes from the biological neural population applying the Levy dynamics for sampling.
This differs from the Langevin dynamics adopted by existing SGMs.
We note that our model aims to mimic the large jump and small roaming functions of the Levy dynamics,
rather than simulating the Levy dynamics via approximating the Levy distribution.
This is because, calculating its Riesz fractional derivative of the Levy dynamics is intractable, as discussed in Sec.3.1

+ As mentioned in Sec. 4.2,
as a functionally similar implementation of the Levy dynamics, our model provides new insights into how to develop SGMs from the following perspectives:
(a) Robustness training: the ability to recover the noises that are outliers of the standard Gaussian distribution, resulting in better sampling quality.
(b) Ensemble learning: Inferring the noises to be denoised by the probabilistic graphical model of empirical Bayes, vs. maximum a posterior by vanilla SGMs.
(c) Multiple paths sampling: the ability to actively adjust the step size during sampling by selecting from $2^T$ different sampling paths.

With our detailed responses, we sincerely appreciate that the reviewers could kindly further review this work and interact with us the authors for any remaining issues and updates.

---

### Author Response · Authors · 2022-11-16
**General Responses to All Reviewers (II)**

We have revised our manuscript with the following updates based on the constructive suggestions from the reviewers:
+ We give a clearer statement about our motivation in the abstract and Sec. 1.

+ We make it clearer that our GM-SGMs do not approximate the Levy distribution, but mimic the functional of the Levy dynamics, i.e., the ability to implement both large jumps and small roaming for better denoising, as revised in Sec. 1 and Sec. 3.2.
+ We emphasize that the sampling process of GM-SGMs is the same as that of original SGMs in Sec. 3.4. This means that our GM-SGMs do not involve any extra complexity during sampling, an advantage over other techniques such as Analytic-DPM that requires extra computation during sampling.
+ We compare our GM-SGMs with relevant methods as suggested by the reviewers as added in A. 4.
+ We add results of the FID under higher iterations in Table 1 and Table 2 as suggested by the reviewers.

 We sincerely appreciate that the reviewers could review our improved manuscript and interact with us for any comments and suggestions.

---

### Decision · Program_Chairs · 2023-01-20

**Decision:**

Reject

**Justification For Why Not Higher Score:**

Although the proposed idea of approximating the Levy dynamics using Gaussian mixtures in diffusion models seems interesting and novel, the reviewers have raised concerns regarding the key motivations of this submission, the quality of approximations, comparisons against accelerated sampling techniques, and the impact of multiple noise scales per time step on the robustness of the score model. Given these concerns, the paper does not appear to be ready for publication at ICLR.

**Justification For Why Not Lower Score:**

N/A

**Metareview: Summary, Strengths And Weaknesses:**

Inspired by biological systems, this paper studies heavy-tailed Levy dynamics for diffusion models to encourage both large jumps and small roaming during sample generation. However, since simulating Levy dynamics is computationally expensive and intractable, they are approximated using Gaussian mixture distributions with shared mean and different variances in this paper.

Pros:
- Introducing new stochastic processes for diffusion models is interesting and novel
- The paper is overall written nicely and clearly. The mathematical notation could be improved slightly as pointed out by vUWR

Cons:
- Demonstrating and motivating why diffusion models need large jumps is missing
- The discussion in Sec 4.2 does not provide theoretical justification for why the proposed method is more robust. Additionally, this analysis does not show that the generative score model actually will decide on large jumps vs small roaming as the final model will be the average of all denoising directions across all possible sampled noise scales (betas) given an x_t.
- The approximation of Levy dynamics with a Gaussian mixture is not theoretically analyzed
- Comparisons against accelerated sampling techniques from diffusion models (both ODE and SDE solvers) would strengthen the position of this paper

**Summary Of Ac-Reviewer Meeting:**

N/A